# Microbial spatial footprint as a driver of soil carbon stabilization

A.N. Kravchenko[1,2,3], A.K. Guber[1,2], B.S. Razavi[4], J. Koestel[5], M.Y. Quigley [1], G.P. Robertson [1,2,6] & Y. Kuzyakov[3,7,8]

Increasing the potential of soil to store carbon (C) is an acknowledged and emphasized strategy for capturing atmospheric $CO_2$. Well-recognized approaches for soil C accretion include reducing soil disturbance, increasing plant biomass inputs, and enhancing plant diversity. Yet experimental evidence often fails to support anticipated C gains, suggesting that our integrated understanding of soil C accretion remains insufficient. Here we use a unique combination of X-ray micro-tomography and micro-scale enzyme mapping to demonstrate for the first time that plant-stimulated soil pore formation appears to be a major, hitherto unrecognized, determinant of whether new C inputs are stored or lost to the atmosphere. Unlike monocultures, diverse plant communities favor the development of 30–150 μm pores. Such pores are the micro-environments associated with higher enzyme activities, and greater abundance of such pores translates into a greater spatial footprint that microorganisms make on the soil and consequently soil C storage capacity.

---

[1] Department of Plant, Soil and Microbial Sciences, Michigan State University, East Lansing, MI 48824, USA. [2] DOE Great Lakes Bioenergy Research Center, Michigan State University, East Lansing, MI, USA. [3] Department of Agricultural Soil Science, University of Göttingen, Göttingen, Germany. [4] Department of Soil and Plant Microbiome, Institute of Phytopathology, Christian-Albrecht-University of Kiel, Kiel, Germany. [5] Swedish University of Agricultural Sciences, Uppsala, Sweden. [6] W. K. Kellogg Biological Station, Michigan State University, Hickory Corners, MI 49060, USA. [7] Institute of Physicochemical and Biological Problems in Soil Science, 142290 Pushchino, Russia. [8] RUDN University, Moscow, Russia. Correspondence and requests for materials should be addressed to A.N.K. (email: kravche1@msu.edu)

Soil C is crucial for soil fertility, and the recovery of some portion of the 133 Pg C lost upon agricultural land-use conversion[1] is a prominent strategy for helping to keep global temperature rises below 1.5 °C[2,3]. Various land management practices are known to build soil C. These practices include reducing soil disturbance[4,5], increasing quantity of plant biomass added to soil[6], and increasing plant diversity[7,8], and perenniality[9,10]. All aim at the two key components necessary to achieve C gains: boosting new C addtions[11,12] and enhancing physical protection of the newly added C[11-13].

However, abundant unexplained examples of these practices working in some but not other circumstances signal that our understanding is incomplete[14]. For example, row crop systems with cover crops can accumulate C even in presence of aggressive tillage[15,16], and some perennial bioenergy systems such as switchgrass (Panicum virgatum) can be slow to increase soil C despite large below ground C inputs and the lack of soil disturbance[9,17]. Such discrepancies suggest a substantial knowledge gap with respect to the plant-soil-microbial interactions that drive soil C accretion. This deficiency is unfortunate, since strategies for effective soil C accrual can only be efficiently developed with a full understanding of the underlying controlling mechanisms.

Whether soil C will be gained or lost stems from the balance between microbial decomposition of new C inputs and protection of new C, either as plant-derived or microbially processed compounds[18], within the soil matrix. Majority of the fresh organic matter inputs comes from plants, i.e., above ground litter and especially roots[19,20], and is subsequently utilized by the microbial community to produce $CO_2$ and decomposition intermediates[21,22]. Intermediate products can escape further decomposition by forming organo-mineral associations with soil mineral surfaces, or by entrapment in small soil pores where they become inaccessible to microorganisms and their exoenzymes[23,24]. Such aspects of soil C balance as sizes and chemical composition of C inputs[19,20], their microbial processing[25], and strength of the physical protection due to soil textural[26,27] and structural[28,29] characteristics have been extensively studied and well understood. Yet, a crucial element kept escaping the research focus: the need for new C to be transported from the micro-sites of its entry and initial processing to the micro-sites where it can be protected from further decomposition. Most studies implicitly assume that C processing and protection are spatially collocated. We suggest that processing and protection take place in different micro-sites of the soil matrix. The need for the new C to be transported to the protective sites prior to protection to occur is the currently overlooked underlying cause for the mentioned above discrepancies in expected and realized C gains.

Transport in soil occurs via pores, which are conduits for gases, water, nutrients, and dissolved organic C, and are habitats for microorganisms[30,31]. Here we provide novel evidence that development of pores of certain sizes can have a controlling effect on C accrual. Pores in the 30–150 μm radius size range may be especially important[32] because they function as likely routes of $O_2$ influx[33,34] and localities of new C inputs from fine plant roots[35]. Their special role is evidenced by greater microbial activity[36–39], higher abundance of diverse taxa[40], and presence of dissolved organic matter enriched in lipids and depleted in lignin[41,42].

We hypothesize that the abundant presence of such pores, along with ample, fresh C inputs and able microbial communities, is necessary for prompt C accrual. We posit that high abundance of such pores contribute to C accrual in a multi-prong fashion. Specifically, they influence influx of fresh C inputs, affect activities of microbial decomposers by modifying their habitats, and optimize conditions for transport of microbial decomposition products into protective soil matrix. Non-invasive X-ray computed micro-tomography (μCT) scanning supplies detailed 3D information regarding the material density distribution within intact soil samples. Hence it is very well suited to outline the locations and geometry of soil pores, promoting, in turn, understanding of chemical and biological processes taking place within the soil pore space[43,44]. We studied soils from five replicated bioenergy cropping systems, established in 2008, to represent a gradient of plant biodiversity[45]. The five systems consist of two monocultures differing in perenniality, namely continuous corn (Zea mays L.) and switchgrass, a two-species system of continuous corn with a winter rye (Secale cereale L.) cover crop, and two systems with high plant diversity, i.e., poplars (Populus sp. hybrid) with an herbaceous understory and a native successional community abandoned from agriculture in 2008. We evaluated cropping system effects on the soil volume in the vicinity of ≥30 μm radii pores, which serves as an indicator of the size of the spatial influence generated by the microorganisms residing in such pores. Using the new approach of in situ zymography we assessed micro-scale spatial patterns in the distribution of six extracellular enzymes representing microbial activities related to C, nitrogen (N), and phosphorus (P) cycling. Then, combining information from 2D zymography and 3D μCT scanning, we localized associations between microbial substrate utilization, as reflected in enzyme activities, and the abundance of soil pores.

Our results suggest that pores with 30–150 μm radii contain the most active microorganisms capable of responding rapidly to fresh C inputs and thus, of producing the greatest amounts of microbially processed C products. These products are important contributors to soil organic matter accumulation[13,46]. Once they move into the adjacent soil matrix, they stabilize there via various physico-chemical mechanisms. The greater the presence of 30–150 μm pores and their spread throughout the soil, the greater the volume of the soil matrix receiving and protecting the products of microbial decomposition, and subsequently the higher soil C accrual. Greater plant diversity stimulates development of 30–150 μm pores and such stimulation is a previously unknown mechanism by which ecosystems with diverse plant communities enhance soil C accrual.

## Results

**Effects of cropping systems on soil C accrual and soil pores**. We collected disturbed soil samples and undisturbed samples, referred to as intact cores, at a replicated field experiment in southwest Michigan, USA. The studied cropping systems were in-place for 9 years, an experimental duration regarded as typically sufficient for detecting soil C changes[47]. Intact soil cores were subjected to X-ray μCT scanning, followed by pore characterization from 3D images (Supplementary Fig. 1).

Consistent with previous reports from this site[9], of the five cropping systems only the two systems with high plant diversity, poplar and native succession, resulted in higher levels of total soil C (Fig. 1a). Lower C levels were observed in continuous corn with and without cover crops and in the switchgrass system. Short-term respiration, an indicator of labile C, followed the same pattern as the soil organic C and was lower in corn and switchgrass than in the native succession system (Supplementary Fig. 2). However, microbial biomass C in switchgrass system was significantly higher than that of continuous corn and not different from that of poplar and native succession systems (Fig. 1a). Belowground primary productivity of the two perennial herbaceous systems, switchgrass and native succession, was similar at the start of the experiment in 2009; however, 3–4 years later, monoculture switchgrass exceeded native succession by 10–20 fold (Fig. 1b).

We assessed the volume of the soil matrix that was in close proximity to ≥30 μm pores (Fig. 2a), that is, within <180 μm

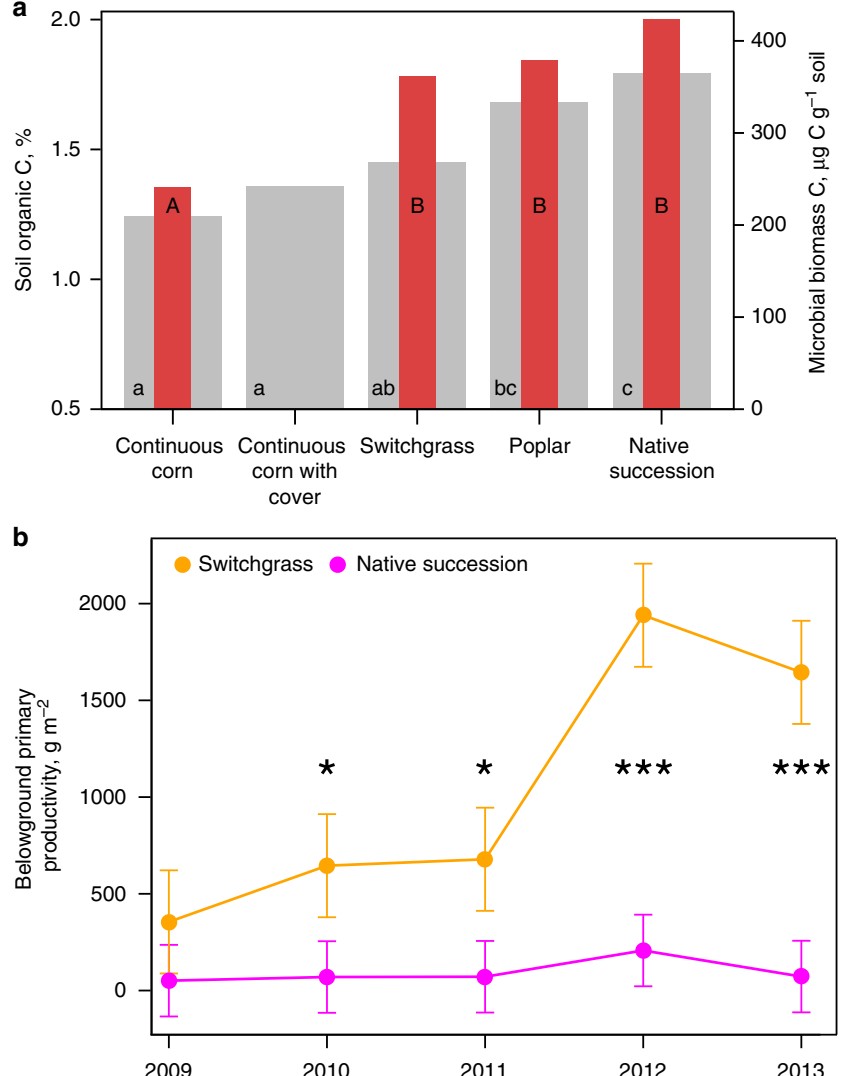

**Fig. 1** Soil organic C, microbial biomass C, and belowground primary productivity in soil of the studied cropping systems 9 years after their implementation. **a** Average soil organic C (gray) and microbial biomass C (red) for five studied systems (5–10 cm soil depth). Lower and upper case letters represent statistically significant differences in terms of soil organic C and microbial biomass C, respectively ($p < 0.05$). Standard errors are equal to 0.1% and 28.5 µg C g$^{-1}$ for soil organic C and microbial biomass C, respectively. Note that microbial biomass C could not be analyzed in the continuous corn with cover crop system. **b** Average belowground primary productivity for switchgrass and native succession vegetation systems from 2009 (the year of establishment) till 2013 (0–10 cm soil depth). Error bars represent s.e.m. Years when switchgrass belowground primary productivity exceeded that of native succession vegetation are marked with *$p < 0.1$ or ***$p < 0.01$. Source data are provided as a Source Data file

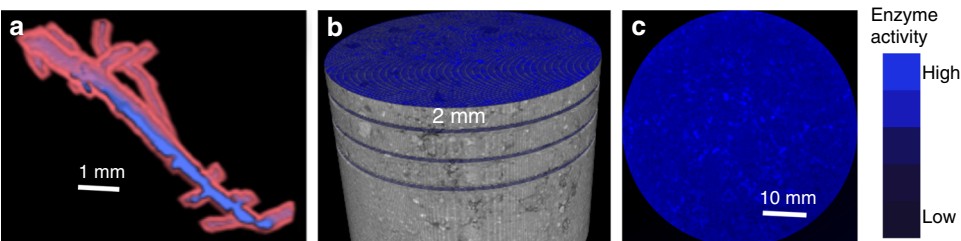

**Fig. 2** Illustration of the steps involved in pore characterization and in combining 2D zymography and 3D X-ray µCT scanning. **a** The volume of the soil matrix (pink) within 180 µm distance from ≥30 µm pores (blue) was used as an approximate indicator of the size of the soil matrix that can be potentially affected by C processing taking place within pores. **b** After µCT scanning (at 30 µm resolution) each soil core was cut into slices, with 2 mm distance between the slices. Each core contained 8–16 slices, and, in most of the cores, zymograms of each enzyme were obtained on two slices per core. **c** A membrane saturated with an enzyme-specific substrate was placed on the surface of each slice and a zymogram was obtained

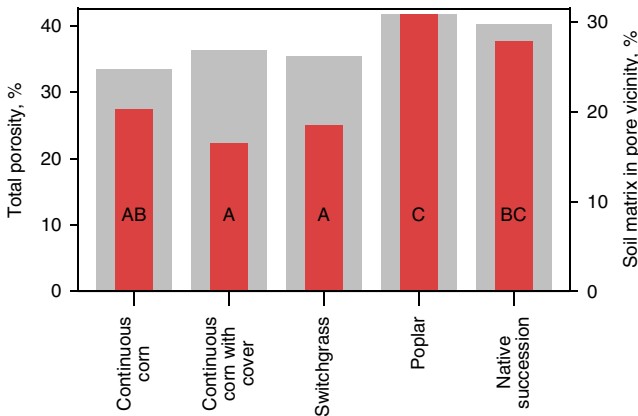

**Fig. 3** Soil pore characteristics of the studied cropping systems 9 years after their implementation. Total porosity (gray bars, left y-axis) and percent of soil matrix located within <180 μm distance from the nearest ≥30 μm pore (red bars, right y-axis). Letters mark significant differences among the systems in terms of percent of soil matrix located within <180 μm distance from the nearest ≥30 μm pore ($p < 0.05$). Source data are provided as a Source Data file

distance from the pore boundaries. The 180 μm distance was used as an approximate indicator of a soil volume potentially affected by processes taking place within the pores and its value was selected as consistent with spatial patterns in microbial and C distributions observed in soil matrixes at similar experimental scales[30,48]. A much smaller portion of the soil matrix in continuous corn and switchgrass was in the vicinity of these pores as compared to poplar and successional soils (Fig. 3). More specifically, the proportion of the soil matrix located close to such pores in native succession and poplar systems exceeded that in switchgrass by 51% and 67%, respectively. Total porosity in the poplar and native succession systems differed considerably from the others (Fig. 3). Since prior to 2008 the experimental site belonged to an agricultural field with common land use and management history, the observed differences are solely a product of contrasting root activities during the last 9 years.

**Associations between enzyme activities and soil pores**. After μCT scanning, the intact soil cores were subjected to 2D zymography. Combining zymograms from >130 soil slices of intact soil cores with 3D X-ray μCT images provided a unique opportunity to associate enzyme activities with pores of different sizes (Fig. 2b, c).

Spatial associations of enzyme activities with soil pores substantially differed between the poplar and successional systems versus the continuous corn and switchgrass systems (Fig. 4a and Supplementary Table 1). Enzyme activities in soils of successional systems were the same in regions with prevalence of <30 μm and 30–150 μm pores, while in poplars the activities associated with prevalence of <30 μm exceeded those in the regions with prevalence of 30–150 μm pores. In contrast, in both corn systems and in switchgrass, the enzyme activities were much lower in the regions dominated by <30 μm pores than in the regions dominated by 30–150 μm pores. This effect was particularly strong in switchgrass soils, and was observed for all studied enzymes except cellobiohydrolase (Supplementary Fig. 3). In all systems the lowest activities were associated with very large (>180 μm) pores.

Notwithstanding differences in enzyme activities among pore sizes, soil enzyme activities from disturbed samples were not much different among the systems (Supplementary Fig. 4). The systems differed only in acid phosphatase activity, which was the

highest in the native succession soil and the lowest in continuous corn.

**Enzyme activities and soil pores after addition of fresh C**. To address our expectation that 30–150 μm pores harbor active microbial communities capable of processing fresh C inputs, we also conducted zymography measurements after soil cores were incubated with fresh plant litter (Supplementary Fig. 1d). Zymography after incubation of soils with fresh plant litter showed more pronounced contrast in spatial associations between enzyme activities and soil pores compared to soil without fresh C (Fig. 4b). The enzyme activity increased in response to fresh C additions especially in the areas with 30–150 μm pores as compared to those with greater abundance of the smaller pores.

**Discussion**
Nine years of implementing cropping systems with different plant species diversities led to the formation of contrasting soil pore size distributions[49]. The more diverse systems, i.e., poplar and native succession, developed higher soil porosity (Fig. 3) and a greater presence of 60–150 μm pores[49] and had concomitantly higher soil C contents (Fig. 1a). The systems with low plant diversity, i.e., continuous corn with and without cover crops and switchgrass, had no pore increases and did not experience soil C gains. In the latter three systems greater enzyme activities were associated with 30–150 μm pores (Fig. 4a). Yet, soil under switchgrass acquired significantly higher microbial biomass than continuous corn, comparable to that of poplar and native succession systems (Fig. 1a).

The finding that greater presence of pores benefits C accrual does appear paradoxical. However, it actually reflects a previously unrecognized micro-scale spatial interaction between physical and microbiological contributors to soil C sequestration. We posit that in soils with fewer 30–150 μm pores, microbial communities have a smaller spatial footprint, which we define as the volume of the soil matrix that can be immediately influenced by microbial decomposition products. A smaller footprint implies a smaller soil volume for entombing microbial decomposition products[50].

We propose the following conceptual model integrating this mechanism in the existing concept of plant-soil-microbial interactions (Fig. 5): Plant roots are the key agents in formation of soil pore architecture. Pores in 30–150 μm size range are the preferential locations of new C inputs and active microbial communities, where active processing of C inputs takes place. The greater the volume of the soil matrix in contact with such pores, the greater is the potential for microbial decomposition products to be transported to and be protected within the soil matrix inaccessible to microbial decomposers. Plant systems that stimulate creation of such pores enhance opportunities for microbial decomposition products to reach inaccessible soil matrix and thus stimulate C gains. Below we provide the evidence from the current study and published literature in support of the proposed concept.

A significant body of research supports the notion that pores in the range of tens-of-μm play an outsized role in C processing. Pores of this size demonstrate faster organic matter decomposition, greater presence of labile organic compounds, and higher microbial abundances[36–42]. The special role of pores of this size range is driven by two factors. First, they are likely preferential locations of fine roots, typically ~30 μm. Fine roots are covered by root hairs contributing to rhizodeposition and have maximal exudation rates. Stable isotope analysis showed that these pores indeed receive and process large amounts of fresh C inputs[51]. Second, these pores possess optimal micro-environmental conditions in terms of water and oxygen supply to resident microbial

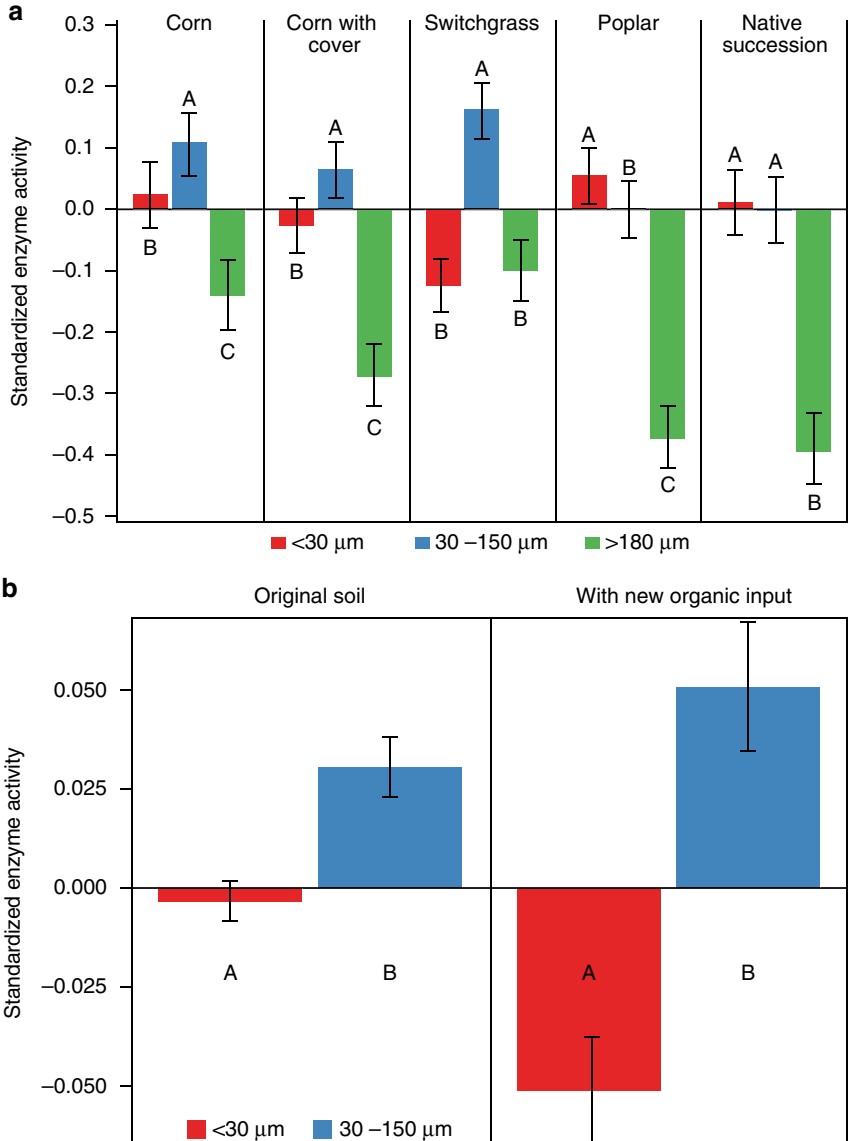

**Fig. 4** Standardized enzyme activities in soil micro-sites with prevalence of pores of three size groups (<30 µm, 30–150 µm, and >180 µm). **a** Enzyme activities from soil slices not subjected to incubations with fresh C inputs. Shown are means across all studied enzymes. Error bars are s.e.m. (based on 139 zymography layers from 13 soil cores). Letters within each cropping system mark statistically significant differences among pore size classes ($p < 0.05$). **b** Enzyme activities from slices subjected and not subjected to incubations with fresh nutrient inputs; shown are means across all systems and enzymes. Error bars are s.e.m. (based on 88 and 14 zymography layers without and with new organic inputs, respectively, from 10 soil cores). Letters mark significant differences between enzyme activities in localities with <30 and 30–150 µm pores ($p < 0.05$). Summary data are provided as a Source Data file

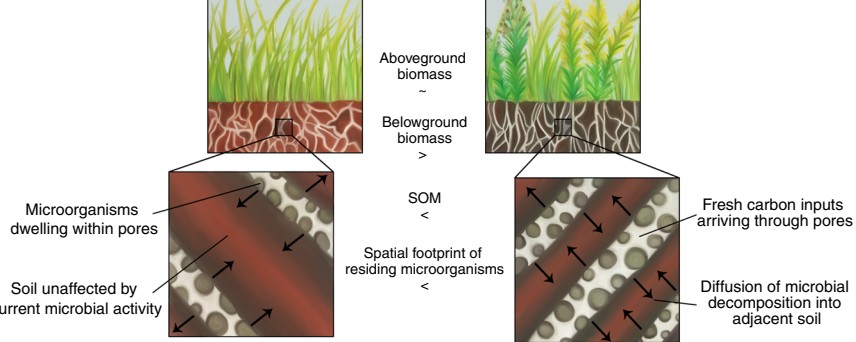

**Fig. 5** Microbial footprint defines the soil volume available for C protection. Schematic representation of the effect that the abundance of 30–150 µm pores has on the size of the spatial footprint of microorganisms residing in such pores in perennial switchgrass monoculture and biodiverse native vegetation systems

communities. Smaller pores tend to have longer periods of anoxia[52] and larger pores tend to have longer periods of water shortage[53]. Higher enzyme activities associated with 30–150 μm pores (Fig. 4) confirm that these pores provide micro-environmental conditions optimal for microbial functioning and also harbor active microbial communities capable of a speedy response to new C inputs.

Our findings suggest that the ability of root systems to generate such pores can be a decisive factor in producing soil C gains. The roots of the two more diverse cropping systems, i.e., poplars and native succession, which quickly accumulated high levels of soil C, apparently stimulated the formation of ≥30 μm pores[49]. Once formation was initiated, these pores became, first, the locations of enhanced microbial activity and intensive processing of new C inputs; and second, the major source of microbially processed C that diffused into and became physico-chemically protected within the surrounding soil matrix by forming associations with mineral surfaces which protected them from further decomposition[11,13]. High abundance of such pores ensured a large proportion of the soil matrix in close proximity to active microbial communities as represented by the percent of soil matrix located within 180 μm of the nearest ≥30 μm pore (Fig. 3).

The switchgrass system with its low abundance of 30–150 μm pores[49], exceptionally high belowground biomass production (Fig. 1b), and a surprising lack of C gains (Fig. 1a; Supplementary Fig. 1) epitomizes the importance of this mechanism. The switchgrass system is very similar to poplar and native succession vegetation in terms of possessing all the components known to be necessary for rapid and extensive soil C accrual. As a perennial system, switchgrass management does not involve annual tillage; its belowground production is massive[54] and greatly exceeds that of the native succession vegetation (Fig. 1b). Switchgrass soils have highly diverse microbial communities, with microbial biomass (Fig. 1a) and bacterial diversity indices greatly exceeding those in the continuous corn system and similar to those in poplar and native succession[55]. Fungal biomass in the switchgrass soils likewise exceeds that of corn and is similar to poplar and native succession[56], and greater fungal abundance is typically associated with greater C accrual[13]. The production of fine roots in switchgrass is also comparable to that of the native succession system[57]. The switchgrass system was in place for nine years, a sufficient time for changes in soil C to take place[47]. Yet even labile soil C fractions, which typically indicate coming changes in total soil C did not increase under switchgrass (Supplementary Fig. 1 and ref. [9]). It should be noted that these results are not unique to the current study. While the conversion of annual cropland to switchgrass often leads to soil C gains[58,59], no or slow increases in soil C concentrations under switchgrass as well as high variability in observed C gains[60] have also been reported, including diverse experimental sites in North Dakota, South Dakota, and Nebraska[61], southeastern US[62], and northern Italy[17].

In soils under switchgrass, high root C inputs[63] stimulated microbial activity, but in the absence of a sufficient abundance of 30–150 μm pores, most of this C was apparently oxidized to $CO_2$ rather than entombed as organo-mineral associations in the nearby soil matrix. While the 30–150 μm pores in switchgrass soil were the sites of intensive enzyme activities and C processing (Fig. 4), the volume of the soil matrix adjacent to such pores was relatively small (Fig. 3). Thus, the pore-adjacent layer likely became quickly saturated with microbial C inputs, with limited diffusion farther from the pores. This is consistent with observations of Chenu et al.[36] who noted that once a new C source was applied to the surface of soil aggregates it was actively microbially consumed in situ with only small amounts diffusing into aggregate centers. A large volume of soil matrix, with high potential for

C physico-chemical protection, therefore remained unutilized, not receiving C, and resulting in overall low soil C levels. Indeed, in switchgrass soil intensive microbial processing and C turnover have been demonstrated in a number of $^{13}C$ isotope based studies[64–67]. It can be surmised that the new C inputs were either fully decomposed and lost as $CO_2$ or lost as dissolved organic C during convective flow events.

Developed differences in soil pore characteristics between monoculture switchgrass and biodiverse poplar and native succession systems could be a result of several physical and biological factors. A potential physical factor is a difference in water usage by plant communities. Systems where soils experience greater drying in summer would be expected to have a greater presence of large pores (cracks) due to soil shrinkage. This phenomenon is commonly observed in soils with high clay contents (2:1), but still, though to a lesser extent, is present in sandy loam soils. Systems with greater soil moisture levels prior to soil freezing in winter would be expected to develop greater pore presence due to more extensive soil expansion during freezing/thawing cycles. However, experimental evidence suggests that in our systems these physical mechanisms could be only minor contributors to the observed pore differences, especially so since the studied soils do not contain significant amounts of shrink-swell clays. Over the years, soil water content under poplars, with its greater pore abundances, tended to differ from that in the other systems (Supplementary Fig. 5). However the observed tendency was actually disadvantageous to pore formation via wetting-drying and freezing-thawing mechanisms mentioned above, because poplar soil tended to be more moist in spring and summer, and drier in fall prior to soil freezing.

Biological factors are thus likely to dominate. Differences in root architecture, including root density, diameter, strength, fine root biomass and mycorrhization, are well known to affect soil porosity[68–70]; and amounts and compositions of root exudates are likewise affected by root characteristics[63]. Systems with greater plant diversity have a greater variety of root architectures with correspondingly different effects on porosity. Moreover, in the systems with no soil disturbance and low plant diversity the reuse of the old root channels by the new plant roots is more likely[71,72], further reducing possibilities for new pore formations.

Overall, our results point to a heretofore missing link in understanding plant-soil-microbial interactions. Plant community composition defines not only the composition of the soil microbial community, but, by affecting presence and characteristics of soil pores, it also defines spatial distribution patterns of regions where the microorganisms can most effectively reside and function as related to C turnover and sequestration. That pattern then defines the spatial footprint that microorganisms can make on the surrounding soil matrix. For soil C processing the size of this footprint translates into the size of the soil volume involved in the protection of microbially-processed C inputs.

Our finding was enabled by comparisons between a monoculture switchgrass and biodiverse native succession and poplar systems– the systems that possessed great similarities in all their features except for pore characteristics. Very high inputs of fresh C in the switchgrass system made the effects of pore presence on microbial functioning especially pronounced and their detection possible using 2D zymography and X-ray μCT tools. We surmise that similar considerations are applicable to the no-till continuous corn systems of this study, where small microbial footprint is probably one of the contributors to their low C levels.

Our work demonstrates how better understanding of microscale interactions between plants, microorganisms, and soil characteristics can have a direct impact on macro-scale (landscape) features of terrestrial ecosystems, i.e., accrual of soil C, and how it can lead to new strategies in sustainable agricultural

management. A potential strategy is the maximization of root diversity as a prerequisite of efficient C accrual, which can be achieved by planting more diverse switchgrass stands, promoting diverse understory in poplar plantations, and intensively using polyculture cover crops in row crop systems.

## Methods

**Field experiment.** The Great Lakes Bioenergy Research Center (GLBRC) experiment had been established in 2008 at Kellogg Biological Station's Long-term Ecological Research Site, Michigan, USA on well-drained Alfisols[73]. Soil texture consists of 65% sand, 27% silt, and 8% clay. The field experiment is a randomized complete block design with five replications[45]. The five bioenergy land use and management system are: no-till continuous corn and no-till continuous corn with winter cover crop of cereal rye, a monoculture switchgrass, a hybrid poplar (*Populus nigra × P. maximowiczii* NM6) with herbaceous understory[9], and an early successional community abandoned from agriculture in 2008. The experimental site was plowed prior to establishment and no further plowing was conducted in either of the systems. All systems were managed using local best agronomic practices[45].

Two intact soil cores (5 cm Ø × 5 cm height) were collected from four replicated plots of each bioenergy system for a total of 40 cores in early spring 2017. The soil cores were taken using a soil core sampler (Soil Moisture Equipment Corp.) into an acrylic cylinder located within the sampler. The sampler had two rings, 5-cm height each, and soil from the first ring was discarded. Thus, the cores were collected precisely from 5–10 cm depth within the soil profile. A disturbed soil sample and an additional soil core for bulk density measurement were also collected from each sampling location. Bulk density was measured for each sampling location using the core method[74] and is reported in ref. [49]. Soil temperatures were around 5 °C at the time of sampling and soil water content levels were around field capacity. During the time period prior to the zymography measurements the cores were kept in a dark at 4 °C, except for times of shipping and during μCT scanning. To preserve the cores during shipping each core was tightly closed on both ends with ridged foil caps and wrapped in several layers of plastic using duct tape. The disturbed samples were stored air-dried.

**X-ray computed μCT scanning and image analysis.** Intact soil cores were subjected to X-ray scanning using a GE Phoenix v|tome|x m scanner at the Institute of Soil and Environment at the Swedish University of Agricultural Sciences in Uppsala, as described in ref. [49]. In brief, the X-ray scanner was equipped with a 240 kV tube, a tungsten target and a 16′′ flat panel detector with 2014 × 2014 detector crystals (GE DRX250RT). Each 3D X-ray μCT image was reconstructed from 2000 projections, with a tube voltage of 130 kV and an electron current of 200 μA. No optical filters were used. Soil cores were scanned inside their acrylic rings. Beam hardening artefacts were not observed. 3D μCT X-ray images were reconstructed using the filtered back-projection approach (GE software datos|x). The entire soil core, including the acrylic cylinder, was scanned. Each image had a voxel size of 29 μm in x, y and z direction.

Two examples of the scanned images are shown in Supplementary Fig. 1a, b. The image processing was conducted using ImageJ/Fiji software. The same image processing procedures were applied to soil samples from all studied systems. Preprocessing consisted of a 3D median filter with a radius of two voxels in all directions using Median 3D filter tool from ImageJ. After that we removed 0.5 cm border parts around each core to avoid artifacts associated with sample wall effects.

For segmenting the images into pores and non-pores we used the minimum error approach[75] on the respective image histograms. The grayscale histograms (8-bit) of all soil cores exhibited a two-distribution pattern, with one distribution corresponding to air + liquid and the other to solid portions of the images. Thus, following ref. [76], the two-Gaussian fits were applied to the histograms. The threshold was computed as a greyscale value that minimized the difference between the overlapping areas of the two distributions. This approach conserves the voxel balance between the two segmented phases (solid and air + liquid). The necessary computations were performed in the Regression Wizard tool of the SigmaPlot software (Systat Software, Inc). The segmentations were conducted separately for each soil core.

Then, visible pores (≥30 μm) were obtained using the Xlib plugin for ImageJ[77]. We used the continuous 3D pore-size distribution option of the software, which provides radii of the largest spheres, that fit into the 3D pore volume, as described in detail in ref. [77]. Therefore, the pore size at a specific location was defined as the radius of the maximally inscribable sphere at this location. Particulate organic matter, including plant roots, in the images was identified as described in ref. [49] and in further analyses particulate organic matter was separated from pore and solid fractions. Pore-size distribution data from the studied cores are reported in ref. [49].

As a measure of the soil matrix fraction that was potentially affected by C processing taking place within the visible pores (≥30 μm), we used the soil matrix volume located within 180 μm distance from these pores (Fig. 2a). The volume was determined via a series of 3D dilations using ImageJ until ~180 μm distance from the pore surface was reached. The value of 180 μm was selected as consistent with

distances for spatial correlations in microbial colony distributions[30] and spatial patterns of soil C distributions[48] in previous micro-scale studies of soil matrix.

**Soil measurements.** The total porosity of each intact soil sample was calculated from its soil volume determined from the μCT image, gravimetric soil moisture content measured at the end of the study, and soil weight. A particle density of 2.6 g cm$^{-3}$ was assumed for the total porosity calculations. Porosity <30 μm was determined as the difference between the total porosity and the image-based porosity, that is, the volume of pores with radii >30 μm, obtained from the images.

For total soil C measurements, disturbed samples were sieved and ground followed by combustion analysis on Carlo-Erba Elemental Analyzer (Costech Analytical Technologies, Valencia, CA). Short-term respiration was determined by incubating 10 g soil samples for 7 days at 20 °C in the dark[78]. Measurements of $CO_2$ emissions were conducted using infrared Photoacoustic Spectroscopy (PAS) (1412 Photoacoustic multi-gas monitors; INNOVA Air Tech Instruments, Ballerup, Denmark) in the gas circulation mode. Microbial biomass C was measured using chloroform fumigation incubation method[79]. After 24 h of fumigation with ethanol-free chloroform, 5 g fumigated and non-fumigated (control) soil samples, were subjected to 10 day incubations followed by $CO_2$ measurements as described above. Microbial biomass C was calculated based on the amounts of $CO_2$ emitted from fumigated and control samples[80].

Gravimetric soil water content monitoring was conducted on a bi-weekly basis from each replicated experimental plot at the GLBRC experimental site. Soil samples for gravimetric soil water content measurements were collected from 0–25 cm depth using a push probe (Ø 2 cm) during growing seasons of 2009–2015. Soil was dried for 48 h at 60 °C.

Root biomass at 0–10 cm depth was measured in soil cores (Ø 7.6 cm) collected in triplicates from each experimental plot. The cores were taken at three points along plant density gradients moving out from the center of a bunchgrass into the interstitial space. Cores were air-dried, sieved over a 2-mm sieve, and roots were washed, dried at 60 °C for 48 h, and weighed. Samples were collected in November from 2009–2013.

The following soil biological characteristics of the studied cropping systems were measured from the studied experimental plots 1–2 years prior to the current work and reported in peer-reviewed literature: microbial community composition and bacterial diversity indices[55], fungal biomass[56], fine root production[57], labile soil C fractions[9].

**2D zymography of soil cores.** Enzyme activities were mapped using soil zymography approach[81]. Zymography mapping consists of placing a membrane saturated with an enzyme-specific fluorogenic substrate on a surface of the soil sample. Once the substrate is enzymatically hydrolyzed, a fluorescent product is generated and the ensuing fluorescent patterns reflect the presence and activities of the enzymes. A photo of the membrane is taken in ultraviolet (UV) light, which then is used to produce a map of enzyme activities on the studied surface, referred to as a zymogram.

A schematic representation of the process of combining zymography with μCT information is shown on Supplementary Fig. 6, while a detailed description of the zymography procedures and soil processing for zymography analyses used in this study are provided in ref. [82]. In brief, the following specific settings were used: membrane—hydrophilic polyamide filters (100 μm thick, Tao Yuan, China)[81]; camera—Nikon D90 camera (Nikon Inc.) with a Sigma 18–250 mm f/3.5–6.3 DC Macro OS HSM lens (Sigma Corp. of America); source of UV light—a 22 W Blue Fluorescent Circline Lamp—FC8T9/BLB/RS (Damar Worldwide 4 LLC); extracellular enzymes—beta-glucosidase, cellobiohydrolase, xylanase, N acetyl-beta-glucosaminidase, leucine aminopeptidase, and acid phosphatase; respective enzyme-specific substrates: 4-Methylumbelliferyl-β-D-Glucoside, 4-Methylumbelliferyl-β-D-Cellobioside, 4-Methylumbelliferyl-β-D-Xylopyranoside, 4-Methylumbelliferyl-N-Acetyl- β-D-Glucosaminide, L-leucine-7-amido-4-methylcoumarin hydrochloride, and 4-methylumbelliferyl-phosphate[83].

Soil cores were cut into 2 mm slices (Fig. 2b) using a specially designed cutting table (Supplementary Fig. 1c), as described in detail in ref. [82]. A total of 13 soil cores were analyzed, with 10–16 zymograms taken per core, one map per each soil slice. Soil moisture levels were not manipulated prior to the measurements. Since good contact between a membrane and a soil surface is needed for obtaining reliable enzyme activity data[84], only portions of each membrane were usable. Such portions were identified via an MUF-staining approach[84] as described by ref. [82]. The areas with minimal contact were excluded from enzyme map analyses.

The zymogram was covered by a 1 mm$^2$ grid, and for each pixel of the grid we identified a corresponding 1 mm$^3$ voxel from the μCT image. The enzyme pixel was assumed to be located at the center of the μCT voxel (Supplementary Fig. 6). For each 1 mm$^2$ pixel of the zymogram we calculated the average grayscale value and standardized it based on the mean and standard deviation of the entire zymogram. For each 1 mm$^3$ voxel of the μCT image we also calculated averages for the studied characteristics, e.g., volumes of pores of different sizes and volume of the soil matrix in relative proximity to sizeable pores. The aggregation of the data to the 1 mm scale conducted here was required to address the uncertainties associated with soil cutting and matching of zymography and μCT images. Because it introduced smoothing into the resulting pore data, to ensure that the uncertainty in pore characteristics of individual voxels was minimized during the analyses of

the relationships between pores and enzyme activities, we only selected for such analyses the voxels where the pore values were above the 95th percentile value for the entire soil core (Supplementary Fig. 7). Specifically, for each core we constructed histograms for pores of all studied class sizes and calculated the 95th percentiles for abundances of each pore size class. Then each voxel of the core was examined and only the voxels with abundances exceeding the 95th percentiles were retained for further analyses. On very rare occasions when a voxel exceeded 95th percentile in more than one pore size class it was grouped in the larger of the size classes. While this approach reduced the number of data points available for the analyses, it ensured that the data used are highly representative in terms of their pore characteristics.

In order to evaluate how addition of new nutrient inputs influences relationships between soil pores and enzyme activities we conducted incubations with fresh plant inputs in 10 studied cores, with 1–2 soil slices per core for a total of 14 slices. Each slice was covered by a layer of nutrient source, either as an alder (*Alnus glutinosa* L.) leaf or as a layer of red clover (*Trifolium pratense* L.) leaves, as shown in Supplementary Fig. 1d. Prior to the experiment, the leaves were air-dried in a botanic press to ensure the maximum uniformity in their contact with the entire soil surface. After placing the dried leaves on the soil surface, 240 µl of water was evenly added with a pipette on top. The surface was covered with aluminum foil and a 100 g sandbag weight to maximize the contact with the soil, and incubated for 3–7 days. After the incubation, the leaves were carefully removed from the surface and an additional 0.2–0.3 mm of soil was cut off the surface to ensure that no leaf material remained on the soil surface subjected to zymography. Then, zymography measurements were conducted on the surface as described earlier.

**Statistical analysis.** Statistical analyses were conducted using PROC MIXED and PROC GLIMMIX procedures of SAS[85]. Implemented statistical models varied depending on the experimental units from which different response variables were collected. Specifically, comparisons among the bioenergy systems in terms of the response variables with a single observation per soil core, e.g., soil C or soil volume in pore vicinity, were conducted using the statistical model with a fixed effect of the bioenergy system and random effects of the experimental field block and experimental field plot (nested within the system). Statistical models for exploring standardized enzyme activities for different enzymes in different bioenergy systems and corresponding to areas with prevailing pores of different sizes included fixed effects of enzyme, system, and pore size class, as well as their interactions, and random effects of blocks, plots, cores (nested within the systems and plots), and also individual soil slices (nested within all of the above factors). Normality of the residuals and homogeneity of variances were checked for each studied variable. In case of marked deviations from normality the data were log-transformed, while in case of variance heterogeneity, unequal variance analysis was performed[86].

## Data availability

Data collected for this study will be made publicly available on Dryad and will be preserved in the KBS LTER database available at http://lter.kbs.msu.edu/datatables. The source data underlying Figs. 1a–b, 2a–b, and Supplementary Figs. 2 and 5 as well as summary data underlying Fig. 4 and Supplementary Figs. 3 and 4 are provided as a Source Data file.

## Code availability

Data and image analyses codes are available from the authors upon request.

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

## Acknowledgements

We are indebted to Maxwell Oerther and Jessica Fry for conducting soil C analyses; and to KBS GLBRC team for agronomic management of the field experiment and for collecting belowground biomass and soil water content data. We would like to thank Chelsea Mamott for the artwork and GLBRC communication team for help with figure preparations. This research was funded in part by the National Science Foundation's

Geobiology and Low Temperature Geochemistry Program (Award# 1630399). Support for this research was also provided by the Great Lakes Bioenergy Research Center, U.S. Department of Energy, Office of Science, Office of Biological and Environmental Research (Awards DE-SC0018409 and DE-FC02-07ER64494), by the National Science Foundation Long-term Ecological Research Program (DEB 1637653) at the Kellogg Biological Station, and by Michigan State University AgBioResearch. The work of A. Kravchenko was supported by DAAD- German Academic Exchange Service' program "Research Stays for University Academics and Scientists, 2017" (57314018) and by the Research Award from Alexander von Humboldt Foundation. We gratefully acknowledge the German Research Foundation (DFG) for supporting the project: RA3062/3-1.

## Author contributions

A.N.K. developed research concepts with inputs from A.K.G, B.S.R., G.P.R., and Y.K.; M.Y.Q. and A.K.G conducted the soil sampling; A.N.K., A.K.G., and B.S.R. conducted laboratory experiments and zymography analyses, with inputs from Y.K.; J.K. and M.Y.Q. conducted computed micro-tomography; A.N.K. and A.K.G. conducted data analyses; A.N.K., G.P.R. and Y.K. wrote the manuscript. All authors contributed to manuscript writing and reviewed the manuscript.

## Additional information

**Competing interests:** The authors declare no competing interests.

