## [Peer Review File · Nature Communications]

Editorial Note: In their review of the third version of this manuscript, reviewer 4 added their comments to the manuscript file. These comments, excluding minor textual revisions, have been copied into this Peer Review File.

Reviewers' comments:

Reviewer #1 (Remarks to the Author):

This paper is extremely well written and laid out quite clearly. I very much like the research, and the innovative link between the tomography and the zymography. However, these are really the only 2 measurements made in this paper, and they are used to support a number of observations that have been made on these research soils elsewhere. I am surprised to not see a C characterization, or a biological assessment (total biomass, substrate-induced respiration) to reflect the microbial component directly. A missing link is that connection between the microbes in these porous habitats and their production of extracellular enzymes. Or, is the pore domain particularly well-suited to preserving enzymes? I would love to see a revision in which the measurements are used more directly.

There is a continuing assumption that microbial processing is required to stabilize C in soil (e.g. Line 86 and elsewhere). It would be useful to clarify that microbial processing is one of the routes to C sequestration. I also think that it merits noting that overall C sequestration/longevity is the balance between microbial processing and C protection. In that event, physical protection is certainly a key mechanism controlling the balance and therefore C residence time. Fundamentally, I think it's important to think about the rate the C enters a soils vs the rate at which it leaves the soil as a key element of C accrual. There's been a lot done in this domain, but not with respect to physical disposition of C. I think your paper agrees with this, yet the comment on line 71 regarding prompt C accrual seems to under-play the decomposition element. Again, this is a comment that could be supported with even a bulk measurement of biomass per unit new carbon (an example).

Line 46: I think treating these mechanisms as a suite of controls on C storage is reasonable. I suspect where we see failures, it is because we don't know how these mechanisms couple under different conditions.

I am intrigued by the proposal to highlight the area within 180 microns of the 30 micron pores, however, I am unclear as to how this value was determined. Additionally, 180 microns linear distance is a very different thing from 180 microns of connected, tortuous pore. I don't know which metric is better, but they authors should be specific about which distance they are using to frame their assumption.

Line 155 describes enzyme values as higher in certain treatments, but the differences were not statistically significant. Please just don't do this – if the values were not statistically significant, the values are considered no different.

Line 194 introduces the zymography, and this is fascinating. I don't see, however, if there was any ability to link the location of fresh carbon, or residual root channels to the most enzymatically active zones. I think this is a key connection to draw, and one that should be discussed.

Where most of the figures are clear and easy to follow, Fig 4a (and SI fig 3) stands out as rather poor, and difficult to follow. I would urge the authors to consider revising this figure to something easier to follow; even panels may help with clarity, though I understand that urge to keep figures simple.

Line 238 introduces a very compelling hypothesis. I think this is a novel consideration – the footprint element – and I think there is a lot to suggest and support this hypothesis. However, I would like to see this more systematically laid out, and the logic of the evidence made more clear. Since there are no direct measurements of a single pore (no criticism, as I know this is near-impossible with current technologies), the inference assembled as evidence needs to be very clear. For example, it would be useful to discuss the typical size of a soil microbe. How many microbes inhabit such pores, or is it a matter of the stabilization of extracellular enzymes.

Line 257 mentions C stabilized with sesquioxides, but these are not directly measured in this research. This is an inference of what could be happening to the carbon, given what we already know about carbon in soils. The assurance that this is relevant in this particular study is weak.

I very much like figure 5. I would, however, suggest that the principle figures of the cropping systems are over-weighted in size. I would make these smaller than the blow-outs, which is where the real concept is presented. I would also suggest integrating some representation of the varied pore sizes and connectivities.

There is discussion of the role of environmental conditions in remodeling soil pore distributions (line 314). As this is presented, it would be useful to refer back to the primary research being presented. Are these soils high in shrink-swell clays? Given the location in Michigan, they must freeze – does soil water content in winter vary with the treatment, in a way that may change the impact of the freeze-thaw?

One observation: the reference list needs to be scrutinized carefully. Several references are cited more than once. I suspect that there is a citation manager glitch, from multiple authors?

Reviewer #2 (Remarks to the Author):

I think the idea behind this study is sound and very interesting, but unfortunately the X-ray CT resolution used in the study means the conclusions are not fully supported by the data. Thus I recommend that this paper be rejected.

The main conclusions of the study is the identification that 30-150 micron soil pores are the most important ones for C. This was done by combining X-ray CT imaging to detect soil pore space and

zymography. However, the X-ray CT scanning was done with a pixel size of 30 microns. This limits the ability to properly characterise pores to significantly bigger than the pixel size. Personally I would say the limit on good characterisation is 10 times the pixel size however it is certainly bigger than 30 microns and 150 microns is only 5 pixels across the pore. The fact that volume is length cubed means that the volume of the pore is even more sensitive to errors in the linear dimensions. The authors are too keen to draw conclusions from the data and haven't fully considered this issue. For example Figures 2 and 4 are not reliable since they are focused on difference in features in the 30-180 micron range.

Reviewer #3 (Remarks to the Author):

The authors have conducted a very interesting study in which they used x-ray microtomography to characterize pore sizes distribution in undisturbed soil samples from a longer-term field experiment and combined this with zymography on selected slices of these samples. They clearly show that a more diverse vegetational cover increases SOC content on a mass basis and enhances soil porosity, especially of the pore classes 30-150 μm . According to their analysis of the zymography data, enzyme activities are also higher in soil compartments containing more pores of this size class, leading them to conclude that a higher microbial activity in these soil pores produces more metabolites that diffuse into the soil matrix and are stabilized there. This is surely a very interesting study that provides important evidence for the role of plant roots and plant diversity in modifying soil physical and biological properties. However, I have some concerns about the data processing and the interpretations of the authors and I would therefore ask the authors to also consider the following aspects in their data analysis and their discussion:

1. Clearly, more diverse plant communities lead to an increase in porosity, from about 33% in the control to about 37-42% in the two most diverse plant communities (Fig. 2a). As a result, bulk density will be considerably lower in the soils with the highest porosity. Therefore, SOC content needs to be given and evaluated on a volume or area basis in order to do correct comparisons between the treatments. Possibly, differences will not be significant, then.

2. The enzyme data is grouped according the "prevalence" of certain pore size classes in the respective voxels. To me it is completely unclear, how this "prevalence" was calculated and if this is indeed meaningful. Does this mean, that a voxel with 1.2% pores $< 30\mu\text{m}$ and 1.1% pores of $150\mu\text{m}$ will be considered in the $< 30\mu\text{m}$ class and if the percentage is the other way around, it would be classified into the larger pore class? This needs to be explained in more detail.

3. The authors argue that microbial products are known for their high sequestration potential (line 87-88) and that they are stabilized in the soil matrix. However, this is largely based on the MEMS theory by Cotrufo et al. (2013) who provide no own experimental evidence for this but developed these ideas from reviewing other studies. The authors should therefore discuss this process more critically. Once the authors include these aspects the manuscript will be acceptable for publication and it will surely be highly recognized in the "SOM community".

Please also make the following changes:

- Line 123, 126: include soil depth of sampling in the figure caption.

- Fig. 3 is more of a methodological illustration and belongs into the supplementary information. Instead, some illustration on how the pore size prevalence was calculated for the voxels would be much more interesting.
- Fig. 2: the parameter “connectivity” seems to be most affected by the treatments, but its meaning and possible relation to enzyme activities and root architecture is not discussed at all. Either do this, or delete it from the results.
- Fig. 4b: If you want to show the contrast between pore size classes, please arrange figure differently by grouping according to treatment and not by size class. Does the figure show, that in smaller pores, enzyme activities decreased after contact with fresh substrate? Or was it similar as before and this is only a statistical effect from normalization? Please explain the figure!
- Line 326f: Please also discuss a possible temporal effect. Same plants will utilize existing root channels repeatedly because they perfectly match in architecture. This is not true for multi-species communities, where root channels from one species may not fit for another species in the following year so that roots will form new channels.

Reviewers' comments are in **bold**, the responses are in regular font.

Reviewer #1 (Remarks to the Author):

This paper is extremely well written and laid out quite clearly. I very much like the research, and the innovative link between the tomography and the zymography. However, these are really the only 2 measurements made in this paper, and they are used to support a number of observations that have been made on these research soils elsewhere. I am surprised to not see a C characterization, or a biological assessment (total biomass, substrate-induced respiration) to reflect the microbial component directly.

Response: We would like to clarify that the supporting measurements that we are referring to in the manuscript were conducted at exactly the same experimental plots of the studied experimental site. We apologize for not making that sufficiently clear in the original manuscript.

The experimental site from where the soil cores of this study were collected has been intensively studied by several research groups in the past 5 years. Previously reported soil biological characteristics from the same plots/soil depths as the ones used in our study include microbial diversities, aggregate stabilities, plant roots; and references to all these studies are provided in the Discussion section of the manuscript. We have provided additional clarifications in the revised Materials and Methods (lines 445-448 in the revised manuscript).

In the original manuscript we reported data on 1) pore characteristics via computed micro-tomography, 2) relative enzyme activity mapping via zymography, 3) total soil C levels, 4) plant root biomass monitoring, and 5) soil moisture level monitoring. Per reviewer's request, we have now also conducted short-term respiration measurements, which provide indications of the overall microbial activities in a course of short-term C processing, and measurements of the total microbial biomass C. Please note that one of the studied systems, namely, continuous corn with cover crops, has been discontinued since 2017, thus no data were collected for it. The data are reported in the revised manuscript (Fig. 1a and Supplement Fig. 2).

We are grateful for the reviewer's suggestion for conducting these additional measurements. The obtained data on microbial biomass C further supported our conclusion that it is not the overall microbial activity, but its spatial spread that matters for enabling soil C gains.

A missing link is that connection between the microbes in these porous habitats and their production of extracellular enzymes. Or, is the pore domain particularly well-suited to preserving enzymes? I would love to see a revision in which the measurements are used more directly.

Response: The reviewer asks an excellent question! Our past work did indicate that abundance of pores in 30-100 μm size range tend to be correlated with abundances of many microbial groups (Kravchenko et al., 2014), supporting the hypothesis that these pores are a primary microbial habitat. However, the suggestion that this "pore domain particularly well-suited to preserving enzymes" can not be ruled out without experimental evidence. In our understanding, separating the two potential causes, namely 1) the greater microbial activity and enzyme

production vs. 2) the greater enzyme preservation, will require novel and non-trivial experimentation. We believe that the data of our current study provide some support in favor of the 1) cause, but they are insufficient to unequivocally decide between the two causes. To the best of our knowledge the present study is the first attempt to specifically relate enzyme activities and pores of different sizes; and we feel that much more work will be needed to elucidate the roles of soil micro-environmental conditions in production versus preservation of soil extracellular enzymes.

There is a continuing assumption that microbial processing is required to stabilize C in soil (e.g. Line 86 and elsewhere). It would be useful to clarify that microbial processing is one of the routes to C sequestration.

Response: It certainly was not our intention to imply that microbial processing is the only route of C sequestration. However, since it is the route that is most relevant to our study and its hypotheses, we have focused on it. We have revised the manuscript to ensure that we do not claim microbial processing to be the only route of C sequestration and added supporting references (lines 53-69, reference 23 – (Barre et al., 2018)).

I also think that it merits noting that overall C sequestration/longevity is the balance between microbial processing and C protection. In that event, physical protection is certainly a key mechanism controlling the balance and therefore C residence time. Fundamentally, I think it's important to think about the rate the C enters a soils vs the rate at which it leaves the soil as a key element of C accrual. There's been a lot done in this domain, but not with respect to physical disposition of C. I think your paper agrees with this, yet the comment on line 71 regarding prompt C accrual seems to under-play the decomposition element. Again, this is a comment that could be supported with even a bulk measurement of biomass per unit new carbon (an example).

Response: Per reviewer's suggestion, we have considerably revised the Introduction providing a more general view of the C input/protection balance (lines 53-69) and expanded on justification behind the study's hypotheses (lines 76-81).

Line 46: I think treating these mechanisms as a suite of controls on C storage is reasonable. I suspect where we see failures, it is because we don't know how these mechanisms couple under different conditions.

Response: We have clarified our view of how the present study adds to and fits within the current paradigm of soil C sequestration (revised Introduction).

I am intrigued by the proposal to highlight the area within 180 microns of the 30 micron pores, however, I am unclear as to how this value was determined. Additionally, 180 microns linear distance is a very different thing from 180 microns of

connected, tortuous pore. I don't know which metric is better, but they authors should be specific about which distance they are using to frame their assumption.

Response: We provided justification for the selected 180 μm distance (lines 152-156 and lines 419-424). We also added an illustration (Fig. 3a) which, hopefully, clarifies what that volume represents. In essence, the value of 180 μm was selected as consistent with distances for spatial correlations in microbial colony distributions from published works and spatial patterns of soil distributions in previous work of our research group (Quigley et al., 2018). We also added the following cautionary note: "Note that it serves as primarily an approximate measure of the soil volume in pore vicinities and cannot be regarded as an absolute range of the pore effects on the surrounding soil matrix." (lines 424-426)

Line 155 describes enzyme values as higher in certain treatments, but the differences were not statistically significant. Please just don't do this – if the values were not statistically significant, the values are considered no different.

Response: We deleted the statement.

Line 194 introduces the zymography, and this is fascinating. I don't see, however, if there was any ability to link the location of fresh carbon, or residual root channels to the most enzymatically active zones. I think this is a key connection to draw, and one that should be discussed.

Response: Unfortunately, in this experiment it was not possible to link locations of fresh C with enzymatically active zones. The plant material was placed as a uniform layer on the surface of the soil core (Supplement Fig. 1d). It was assumed that the entire soil surface had equal contact with the fresh C supply from the incubated plant leaves. The information about pore characteristics was available for these soil cores and, thus, it was possible to relate it to the enzyme information as was done for the non-incubated cores.

Where most of the figures are clear and easy to follow, Fig 4a (and SI fig 3) stands out as rather poor, and difficult to follow. I would urge the authors to consider revising this figure to something easier to follow; even panels may help with clarity, though I understand that urge to keep figures simple.

Response: We agree with the reviewer that we might have tried to squeeze too much data into this figure. While considering the reviewer's comment we found it to be not possible to improve readability while still retaining all the information. We followed the reviewer's advice and presented Fig. 4a as a set of panels. However, for such presentation to be readable we had to reduce the number of reported pore size classes. Specifically, we joined all pore sizes from 30 to 150 classes into one 30-150 group (also please see the response to Reviewer 2). The statistical analysis was redone according to this new classification. However, we believe that the information on all individual pore size classes that was originally shown in Fig. 4a was of potential interest and importance, thus we reorganized it into a table and placed in the

Supplemental materials (Supplement table 1). We still would like to keep Supplement Fig. 3 in the same format, since we believe that this format is optimal for graphical representation of all the relevant data on different pore size classes and cropping systems, even though it does require an effort on the part of the viewer.

Line 238 introduces a very compelling hypothesis. I think this is a novel consideration – the footprint element – and I think there is a lot to suggest and support this hypothesis. However, I would like to see this more systematically laid out, and the logic of the evidence made more clear. Since there are no direct measurements of a single pore (no criticism, as I know this is near-impossible with current technologies), the inference assembled as evidence needs to be very clear. For example, it would be useful to discuss the typical size of a soil microbe. How many microbes inhabit such pores, or is it a matter of the stabilization of extracellular enzymes.

Response: We have conducted a thorough revision of the Discussion, reorganizing and streamlining hypothesis and supporting evidence presentation as suggested by the reviewer.

Line 257 mentions C stabilized with sesquioxides, but these are not directly measured in this research. This is an inference of what could be happening to the carbon, given what we already know about carbon in soils. The assurance that this is relevant in this particular study is weak.

Response: We have removed the specific mentioning of sesquioxides and replaced it with a more inconspicuous statement of protection due to associations with soil solids/minerals (lines 58-59)

I very much like figure 5. I would, however, suggest that the principle figures of the cropping systems are over-weighted in size. I would make these smaller than the blow-outs, which is where the real concept is presented. I would also suggest integrating some representation of the varied pore sizes and connectivities.

Response: We have modified the figure as suggested by the reviewer.

There is discussion of the role of environmental conditions in remodeling soil pore distributions (line 314). As this is presented, it would be useful to refer back to the primary research being presented. Are these soils high in shrink-swell clays? Given the location in Michigan, they must freeze – does soil water content in winter vary with the treatment, in a way that may change the impact of the freeze-thaw?

Response: Unfortunately, we did not collect soil moisture data during the winter. Our last consistently sampled data are from Octobers, and as mentioned in the Discussion, these data do not suggest that freeze-thaw effects played an important role in the observed pore differences. In fact, they contradict the observed differences in pore characteristics. The observed soil moisture

patterns prior to winter freezing are not conducive to increased pore formation via freeze-thaw. We believe that it is highly unlikely that there will be any differences in the treatment effects on the soil moisture levels that would develop during winter after the plant communities went dormant (that is, after our October samplings). The studied soils are low in shrink-swell clays, and we have indicated it in the revised manuscript (lines 345-346).

One observation: the reference list needs to be scrutinized carefully. Several references are cited more than once. I suspect that there is a citation manager glitch, from multiple authors?

Response: The references were checked and corrected.

Reviewer #2 (Remarks to the Author):

I think the idea behind this study is sound and very interesting, but unfortunately the X-ray CT resolution used in the study means the conclusions are not fully supported by the data. Thus I recommend that this paper be rejected.

The main conclusions of the study is the identification that 30-150 micron soil pores are the most important ones for C. This was done by combining X-ray CT imaging to detect soil pore space and zymography. However, the X-ray CT scanning was done with a pixel size of 30 microns. This limits the ability to properly characterise pores to significantly bigger than the pixel size. Personally I would say the limit on good characterisation is 10 times the pixel size however it is certainly bigger than 30 microns and 150 microns is only 5 pixels across the pore. The fact that volume is length cubed means that the volume of the pore is even more sensitive to errors in the linear dimensions. The authors are too keen to draw conclusions from the data and haven't fully considered this issue. For example Figures 2 and 4 are not reliable since they are focused on difference in features in the 30-180 micron range.

Response: We would like to respectfully disagree with the reviewer's opinion that "**the limit on good characterisation is 10 times the pixel size**". We would like to point that our work is in-line with a very large body of mainstream research that applies computed micro-tomography to analyses of soil pore characteristics. Below is just a small sample of recent publications from several research groups that feature X-ray image-derived pore-size distributions. All of them allow inference to the smallest considered pore size, which was in all cases identical with the image resolution.

1. Bouckaert, L., Sleutel, S., Van Loo, D., Brabant, L., Cnudde, V., Van Hoorebeke, L., De Neve, S. 2013. Carbon mineralisation and pore size classes in undisturbed soil cores. *Soil Research*, **51**(1), 14-22;

2. Rabbi, S.M.F., Daniel, H., Lockwood, P.V., Macdonald, C., Pereg, L., Tighe, M., Wilson, B.R., Young, I.M. 2016. Physical soil architectural traits are functionally linked to carbon decomposition and bacterial diversity. *Scientific reports*, **6**, 33012-33012;
3. Dal Ferro, N., Charrier, P., Morari, F. 2013. Dual-scale micro-CT assessment of soil structure in a long-term fertilization experiment. *Geoderma*, **204**, 84-93;
4. Pagenkemper, S.K., Athmann, M., Uteau, D., Kautz, T., Peth, S., Horn, R. 2015. The effect of earthworm activity on soil bioporosity - Investigated with X-ray computed tomography and endoscopy. *Soil & Tillage Research*, **146**, 79-88;
5. Bird, M.I., Ascough, P.L., Young, I.M., Wood, C.V., Scott, A.C. 2008. X-ray microtomographic imaging of charcoal. *Journal of Archaeological Science*, **35**(10), 2698-2706.

Still, regardless of the commonly used current practices, we would like to provide a more rigorous argumentation on why we believe the approach we used in analyses of relationships between enzymes and pores in this study is valid and meaningful:

The reviewer's statement regarding the limit on good characterization might be correct when the goal of the image analysis is exact shapes and characteristics of individual objects, e.g. a study of a single specific sphere (see (Kaestner et al., 2017)). However, in analyses of pore-size distributions it is the average of a multitude of objects that is being considered. Thus, over- and underestimations that stem from the discrete representation of the imaged objects significantly cancel each other out. In the Appendix below we provide a detailed illustration of this effect using simulated images containing spheres with various diameters and coordinates. The simulations indicate that the estimation errors for the averages, while large for the objects with radius equal to the resolution, become quite small for the objects with radii only two times the resolution. This argument is of the same nature as the basic approach to confidence intervals in linear regression predictions - the confidence interval for predicting a sub-population average is much more narrow and the prediction is more accurate than that for predicting an individual value.

We fully agree with the reviewer that exact volumes of pores with sizes close to image resolution can be quantified incorrectly, specifically, underestimated. However, again, we believe that the "**10 times the pixel size**" limit suggested by the reviewer is somewhat exaggerated. Studies that addressed the issue of effects of scanning resolution on pore-size distribution estimations demonstrated that there is no need for concern regarding pores with diameters >3 times the resolution (Vogel et al., 2010; Schluter et al., 2011). We have conducted another simulation to demonstrate the effect of a median filter (as applied in this study) on estimating pore volumes in the studied samples (please see Appendix below). Consistent with the listed above published works, the simulation indicates that underestimation is only the cause for concern for pores with diameters ≤ 2 times the resolution. Thus, in our original images it only affects the pores with radii $30 \mu\text{m}$ ($60 \mu\text{m}$ diameter), and in the median-filtered images used in the final analyses of our study it affects pores with radii 30 and $60 \mu\text{m}$ (60 and $120 \mu\text{m}$ diameters).

However, we would like to point out that due to the nature of image processing, matching the μ CT images with zymograms, and selecting data for analyses of pore/enzyme associations the concerns with underestimations of the small pore volumes become rather immaterial. Specifically, as described in the Methods section, both μ CT scans and zymograms were aggregated on a 1 mm^3 basis prior to extracting pore volumes and relative enzyme activities. After that, for each studied pore size class we only selected the 1 mm^3 cubes where presence of the studied pores was the highest possible, that is, greater than the 95th percentile of the studied pore size class in the entire image (Supplement Fig. 6). Thus, even with underestimation of the presence of the pores of small size classes, we still used in our analyses of enzyme activities only those sections where the pores of these classes were indeed present in large quantities. It is possible that, some of the large pores were misclassified as smaller ones and contributed to some of the respective >95th percentile cubes. However, we believe that the role of such misclassification became progressively smaller, since pores of larger size classes were less, if at all, underestimated.

Nevertheless, to reduce the effect of this additional uncertainty on presentation of our results we modified Fig. 2b by excluding from it the data for $30 \text{ }\mu\text{m}$ radius pores, as those that are most affected by the underestimation problems. We also modified Fig. 4a by combining the data for $30\text{-}150 \text{ }\mu\text{m}$ radii pores into one size class to reduce uncertainties associated with estimations of the individual pore sizes. We also added a cautionary statement to Materials and Methods pointing to the underestimation issue (lines 416-418): "It should be noted that volumes of pores with diameters at or ≤ 3 times image resolution tend to be underestimated during μ CT image analyses (Vogel et al., 2010; Schlueter et al., 2011)."

The only component of the analysis that could have been materially affected by the small pore underestimation is the enzyme activities in soil areas with no visible pores ($<30 \text{ }\mu\text{m}$). Since $30\text{-}90 \text{ }\mu\text{m}$ radii pores actually might have been present in such areas, the enzyme activities that we obtained for the areas with no visible pores could be overestimated. However, this suggests that our observations of lower enzyme activities in the areas with no visible pores in corn and switchgrass treatments are the conservative estimates. With greater accuracy of $30\text{-}90 \text{ }\mu\text{m}$ pore detection, the enzyme activities in such areas would likely be even lower than the currently observed.

Reviewer #3 (Remarks to the Author):

Kravchenko et al. (Nature communications NCOMMS-18-22189) The authors have conducted a very interesting study in which they used x-ray microtomography to characterize pore sizes distribution in undisturbed soil samples from a longer-term field experiment and combined this with zymography on selected slices of these samples. They clearly show that a more diverse vegetational cover increases SOC content on a mass basis and enhances soil porosity, especially of the pore classes $30\text{-}150 \text{ }\mu\text{m}$. According to their analysis of the zymography data, enzyme activities are also higher in soil compartments containing more pores of this size class, leading them to conclude that a higher microbial activity in these soil pores produces more metabolites that diffuse into the soil matrix and are stabilized there. This is surely a very interesting study that provides important evidence for the role of plant roots and plant diversity in modifying soil physical and biological properties.

However, I have some concerns about the data processing and the interpretations of the authors and I would therefore ask the authors to also consider the following aspects in their data analysis and their discussion:

1. Clearly, more diverse plant communities lead to an increase in porosity, from about 33% in the control to about 37-42% in the two most diverse plant communities (Fig. 2a). As a result, bulk density will be considerably lower in the soils with the highest porosity. Therefore, SOC content needs to be given and evaluated on a volume or area basis in order to do correct comparisons between the treatments. Possibly, differences will not be significant, then.

Response: We would like to respectfully disagree with the reviewer regarding the need for this work to present the differences between the studied systems on a volume or area basis.

Reporting SOC stocks is, indeed, important for assessing the changes in the overall C levels in soil profiles in response to land use and management differences. Its correct representation requires not only accounting for changes in the bulk density, but also for respective changes in soil depth, and is most useful when conducted for the entire soil profile. We do agree with the reviewer that statistical significance levels of C stock and C concentration results can differ. But the reason for that is not only the process-driven concomitant increases in soil porosity along with SOC concentrations, but also a purely technical effect of error propagation due to multiplying several measured variables (i.e., concentration, porosity, depth) with their own measurement errors. Thus, to us, assessing SOC stocks and their changes appear to be not a trivial matter but a task that requires special research focus and, thus, has to be warranted by the study's objectives.

The objective of the current study is not to detect changes in SOC stocks of these systems, but to better understand the processes leading to soil C protection/gains within the studied cropping systems. We believe that looking at changes in SOC concentrations and changes in soil pore architectures separately, as done here, is more instrumental and appropriate for our study goals.

2. The enzyme data is grouped according the “prevalence” of certain pore size classes in the respective voxels. To me it is unclear, how this “prevalence” was calculated and if this is indeed meaningful. Does this mean, that a voxel with 1.2% pores < 30µm and 1.1% pores of 150 µm will be considered in the < 30 µm class and if the percentage is the other way around, it would be classified into the larger pore class? This needs to be explained in more detail.

Response: We have provided detailed explanations of how prevalence of pores of different sizes was determined (lines 516-527). We also added a figure that illustrated the approach (Supplement Fig. 6).

3. The authors argue that microbial products are known for their high sequestration potential (line 87-88) and that they are stabilized in the soil matrix. However, this is largely based on the MEMS theory by Cotrufo et al. (2013) who provide no own experimental evidence for this but developed these ideas from reviewing other studies. The authors should therefore discuss this process more critically.

Response: We modified the statement so that it does not focus on the questionable "sequestration potential" aspect but on well accepted concept of microbial products forming the basis of SOM (lines 98-99).

Once the authors include these aspects the manuscript will be acceptable for publication and it will surely be highly recognized in the "SOM community".

Please also make the following changes:

- **Line 123, 126: include soil depth of sampling in the figure caption.** •

Response: Added as requested (lines 127 and 133).

Fig. 3 is more of a methodological illustration and belongs into the supplementary information. Instead, some illustration on how the pore size prevalence was calculated for the voxels would be much more interesting.

Response: We would like to respectfully disagree with the reviewer. While Fig.3 is indeed a methodological illustration we believe it is important for understanding of how the key data of the manuscript were obtained. Many readers might not be interested in downloading the supplementary materials and thus, without this figure, will likely have difficulties with following how the data were obtained from the main text only. As suggested by the first reviewer, the concept of a soil volume around the pores might also be difficult to follow, thus we added a visualization of that to Fig.3. As mentioned earlier, we have now included a detailed description of pore size prevalence calculations in the Materials and Methods and have a figure (Supplementary Fig.6) that illustrates that concept.

- **Fig. 2: the parameter "connectivity" seems to be most affected by the treatments, but its meaning and possible relation to enzyme activities and root architecture is not discussed at all. Either do this, or delete it from the results.**

Response: As suggested by the reviewer we removed the connectivity results.

- **Fig. 4b: If you want to show the contrast between pore size classes, please arrange figure differently by grouping according to treatment and not by size class. Does the figure show, that in smaller pores, enzyme activities decreased after contact with fresh substrate? Or was it similar as before and this is only a statistical effect from normalization? Please explain the figure!**

Response: As suggested, we rearranged Fig. 4b by grouping by type of incubation instead of pore size classes. The enzyme readings were standardized for each individual zymography layers in each soil core. Thus the lower standardized enzyme activities in localities with prevalence of very small pores after incubation with fresh plant leaves might to some extent reflect the standardization - that is, an increase in the spatial contrast in enzyme activities.

- **Line 326f: Please also discuss a possible temporal effect. Same plants will utilize existing root channels repeatedly because they perfectly match in architecture. This is not true for multi-species**

communities, where root channels from one species may not fitfit for another species in the following year so that roots will form new channels.

Response: The reviewer made an excellent point! We have included that in the discussion (lines 355-357)

Appendix: two simulated illustrations to address concerns of Reviewer 2

We are grateful to the reviewer for drawing our attention to this issue. We agree in-so-far with the reviewer that the porosity for the pore size classes close to the image resolution are associated with large errors and we provided a more careful discussion of this aspect in the revised manuscript.

We strongly disagree with the statement that a ratio of 10 between resolution and pore diameter is needed for a proper quantification of the average porosity in X-ray images. The statement is certainly true if individual objects are sought to be analyzed, e.g. one single specific sphere. (see Kaestner, A. P., et al. (2017). Samples to Determine the Resolution of Neutron Radiography and Tomography. *Physics Procedia* 88: 258-265.). However, if the average of a multitude of objects is considered, as in our study, over and underestimations that stem from the discrete representation of the imaged objects cancel out each other significantly. We have demonstrated this by drawing spheres of different radii on a 3-D mesh to test how a voxel discretization of a sphere captures its true value.

Depending on where the center of the sphere is located with respect to the voxel grid, its volume will be over or underestimated given a specific sphere radius. There are two extreme locations possible: 1) the sphere's center coincides with the center of a voxel and 2) the sphere's center is identical with the corner coordinates of 8 neighboring voxels. All other locations represent intermediate cases of these two cases. In case 1), the voxel will be filled as soon the sphere's volume exceeds half of the voxel's volume, i.e. for sphere radii of > 0.4924 . The volume of a sphere located exactly at a center of a voxel remains undetected if its radius is smaller than 0.4924 (100% relative underestimation of its volume). But its volume is overestimated for slightly larger radii, until its volume becomes larger than the voxel volume, when its volume will be underestimated again, but to a smaller relative amount as previous overestimation. When the first neighboring voxels 'detect' the sphere, its volume will again be overestimated. The error will thus undulate between over and underestimations with the latter becoming smaller and smaller. In case 2), in contrast, the sphere will remain undetected until its volume becomes larger than the one of 4 voxels, i.e. for a radius of 0.9847. Also the error for a volume of such a voxel will undulate around zero with ever smaller relative values. Figure 1 depicts the volume errors of 1000 spheres with random coordinates relative to the voxel grid, a number which is realistic for isolated pores in an X-ray image of the dimensions used in our study. This exercise demonstrates that the average discretization error is less than 5 % even for voxels with radii of approximately the image resolution. For radii of more than two times the image resolution become smaller than 0.5%. Admittedly, this only applies for ideal, error-free images. In our study, we were using our X-ray scanner well within the resolution range it was designed for, which rules out blurring artefacts due to a disproportionate focal spot size. However, the images are affected by a salt-and-pepper type noise. To remove this noise, we applied a 3-D median filter with a radius of 2 voxels. This operation also removes a large fraction of the smallest pore classes. It follows, that the average porosity pertaining to the pores with a diameter close to the image resolution will be severely underestimated. Likewise the number of pores with radii close to the image resolution are strongly underestimated.

Figure 1: Average error in volume estimation of 1000 realisations of spheres with centers at random distance to the voxel center.

We conducted another exercise to demonstrate the error in the pore volume estimation introduced by using a median filter. We used four randomly selected X-ray images from the ones presented in our study. We cut out a 800 x 800 x 800 voxel region of interest for each of them. We then made a copy of each region of interest. For the first copy, we directly applied a manual segmentation based on the image histogram, as in the original manuscript. This segmented image includes all voxel sized pores together with voxel sized artefacts due to image noise, i.e. some spurious extra pore and minus a lesser amount of missing pores. The latter number is smaller, because the probability that the respective noise occurred at a pore location is much smaller than an occurrence in the soil matrix, which makes up the bulk of the soil volume.

To the other copy, we applied a 3-D median filter of radius two, and then conducted the segmentation, using the identical threshold as before. We then carried out an analysis of thickness on all images, using the maximum inscribed sphere method. Finally, we extracted the pore size distribution radii between 30 and 300 micron for all images and compared the pore size distributions of images with and without median filter applied.

This exercise reveals that the underestimation persists for the two smallest resolvable pore-size classes, as Figure 2 depicts. Here the pore volumes are underestimated. For the next largest pore sizes, the pore volume may be over or underestimated, depending on the shape of the pores. Again, if average porosities of several images are considered, the errors introduced by the application of the median filter become small for all pore-size classes equal or larger than three times the image resolution.

Figure 2: Porosities for four randomly selected soil samples from the image dataset presented in our study. On the left, porosities of individual soil samples are shown, either obtained from the unfiltered or from the median-filtered images. On the right, the mean porosities for the filtered and unfiltered images are shown. 'S' stands for sample.

References:

1. Barre, P., Quenea, K., Vidal, A., Cecillon, L., Christensen, B.T., Katterer, T., Macdonald, A., Petit, L., Plante, A.F., van Oort, F., Chenu, C., 2018. Microbial and plant-derived compounds both contribute to persistent soil organic carbon in temperate soils. *Biogeochemistry* 140, 81-92.
2. Kaestner, A.P., Kis, Z., Radebe, M.J., Mannes, D., Hovind, J., Grunzweig, C., Kardjilov, N., Lehmann, E.H., 2017. Samples to determine the resolution of neutron radiography and tomography. *Neutron Imaging for Applications in Industry and Science* 88, 258-265.
3. Kravchenko, A.N., Negassa, W.C., Guber, A.K., Hildebrandt, B., Marsh, T.L., Rivers, M.L., 2014. Intra-aggregate Pore Structure Influences Phylogenetic Composition of Bacterial Community in Macroaggregates. *Soil Science Society of America Journal* 78, 1924-1939.
4. Schluter, S., Weller, U., Vogel, H.J., 2011. Soil-structure development including seasonal dynamics in a long-term fertilization experiment. *Journal of Plant Nutrition and Soil Science* 174, 395-403.
5. Vogel, H.J., Weller, U., Schluter, S., 2010. Quantification of soil structure based on Minkowski functions. *Computers & Geosciences* 36, 1236-1245.

Reviewer #2 (Remarks to the Author):

I am very sorry to write this review, but in my opinion the authors have not addressed the issue of image resolution related to pore size distribution detection and resulting errors in their analysis. It is unfortunate that they only had access to an imaging system that had a spot size 30 microns. Thus, they cannot reliably detect any pores that are comparable to this size. The authors seem to make the case that it shouldn't matter, but the case they are making is unconvincing especially since the papers they cite are soil science papers and not microscopy/stereology papers. Just because somebody (and they reference their own work) else in earlier paper in a different soil science domain journal uses the method does not mean that the method is appropriate for this study and that the science conclusions are valid.

Specifically, the data on Fig 2b should not show 60 micron pore size since this is only two pixels across the pore and hence there is no way of knowing how big the pore actually is i.e. one cannot detect the pore edges with 2 pixels across the pore (even 3 pixels across the pore does not allow for pore size detection). The whole discussion in the response letter about the averaging is not rigorous from the microscopy and stereology point of view. One cannot average inaccurate data, especially when it is so central to the paper's conclusions.

On Fig 4a the authors have pooled the data from 30-150 microns. This is again not scientifically rigorous. They should exclude any data from 30-90 micron range from this pooling as it is not a reliable data. I suspect their science conclusions would have to change if they did this, but this would be technically the right thing to do. The hint to this is in fact on Fig 4b where the largest/significant differences can be seen in <30 micron pore results and we know due to the imaging resolution and method used this is the region in the data where the imaging method is not valid.

I am truly sorry to be so negative. The authors are asking the right science questions, but they are using a method (imaging system resolution) that is not fit to answer the questions they are asking. Most of their findings are for pores 30-150 microns in diameter i.e. 3 pixels across and hence the data at this scale is not reliable as one cannot detect a single interface reliably using this method.

Reviewer #3 (Remarks to the Author):

Although the authors have not fully followed my suggestions and requests for changes to the manuscript, I respect their arguments for not doing this. Overall, the revised manuscript has greatly improved in clarity, quality and legibility. From my point of view I see no need for further revisions and therefore recommend to accept the submission for publication.

Reviewer: I am very sorry to write this review, but in my opinion the authors have not addressed the issue of image resolution related to pore size distribution detection and resulting errors in their analysis. It is unfortunate that they only had access to an imaging system that had a spot size 30 microns. Thus, they cannot reliably detect any pores that are comparable to this size. The authors seem to make the case that it shouldn't matter, but the case they are making is unconvincing especially since the papers they site are soil science papers and not microscopy/stereology papers. Just because somebody (and they reference their own work) else in earlier paper in a different soil science domain journal uses the method does not mean that the method is appropriate for this study and that the science conclusions are valid.

Response: We believe that the statements made above by the Reviewer perfectly pinpoint the misunderstanding problem that we face here. The respected Reviewer appears to be an expert in microscopy/stereology, but he/she is not familiar with X-ray μ CT. Microscopy and μ CT are two very different techniques, and even though both produce images, they are based on very different mathematical reconstruction concepts. The biggest difference is that microscopy is largely a 2D technique; thus stereology is indeed needed to extract 3D information from 2D microscopy images. μ CT scanning produces 3D images; thus no extrapolations from 2D into 3D is necessary. Given that microscopy/stereology has nothing to do with X-ray μ CT, it is obvious that "microscopy/stereology papers" were not cited in our manuscript. We have to say that it would indeed be strange and irrelevant to cite anything on microscopy/stereology in any study that utilizes μ CT in porous media/soils pore-size distribution analyses.

Reviewer: Specifically, the data on Fig 2b should not show 60 micron pore size since this is only two pixels across the pore and hence there is no way of knowing how big the pore actually is i.e. one cannot detect the pore edges with 2 pixels across the pore (even 3 pixels across the pore does not allow for pore size detection). The whole discussion in the response letter about the averaging is not rigorous from the microscopy and stereology point of view. One cannot average inaccurate data, especially when it is so central to the paper's conclusions.

Response: It is now clear to us where this whole problem comes from - the Reviewer is again referring to his/her 2D microscopy/stereology experience. We fully agree that when extrapolating from a 2D image into a 3D, it indeed would not be possible to know how big the

pore that is 2-3 pixels across actually is. But in X-ray μ CT we are not extrapolating from 2D to 3D! We do have the actual 3D images! And we do not even work with pixels - we work with 3D voxels. So we do know the size of the pores that we identify in our 3D images. The issues we have to face in our analyses is that some of these pores might not be detected by our thresholding procedures, thus there might be under/over-estimations in volumes of pores of different sizes. In our previous reply to the comments from this Reviewer we greatly elaborated on a potential effect of these issues on our results, because we thought that this is what the Reviewer's concern is. But it is clear now that we and the Reviewer were talking about two completely different things. Our "... whole discussion in the response letter about the averaging" was not just "not rigorous from the microscopy and stereology point of view", it was irrelevant "from the microscopy and stereology point of view". However, it was very relevant and rigorous from X-ray μ CT point of view. Again, the microscopy and stereology point of view is irrelevant when it comes to determining pore-size distributions of porous materials (such as soils) from 3D μ CT images.

Reviewer: On Fig 4a the authors have pooled the data from 30-150 microns. This is again not scientifically rigorous. They should exclude any data from 30-90 micron range from this pooling as it is not a reliable data. I suspect their science conclusions would have to change if they did this, but this would be technically the right thing to do. The hint to this is in fact on Fig 4b where the largest/significant differences can be seen in <30 micron pore results and we know due to the imaging resolution and method used this is the region in the data where the imaging method is not valid.

Response: Unfortunately, the illustration that the Reviewer provides with Fig. 4b and <30 micron pores makes it again very clear that the Reviewer is indeed not familiar with the nature, analysis, and interpretation of the μ CT data in soils and porous media. The locations with prevalence of <30 micron pores, that is, the locations with prevalence of the pores below the resolution of our 3D images, are NOT "the data where the imaging method is not valid"! These are the voxels that contained no pores visible at the studied resolution. If anything, they are the most reliable part of the pore-size distribution data from μ CT analysis of porous media, since they are least subjected to partial volume effects and other uncertainties.

Reviewer: I am truly sorry to be so negative. The authors are asking the right science questions, but they are using a method (imaging system resolution) that is not fit to answer the questions they are asking. Most of their findings are for pores 30-150 microns in diameter i.e. 3 pixels across and hence the data at this scale is not reliable as one cannot detect a single interface reliably using this method.

Response: We are glad to hear that the reviewer positively assesses our research questions. It is indeed unfortunate that he/she looked at X-ray μ CT scanning in soils/porous media techniques from a microscopy technique perspective.

Reviewers' comments:

Reviewer #4 (Remarks to the Author):

I read the paper carefully and made several comments on it. I believe that this research work is important and present interesting results about the soil porous system and soil carbon stabilization. The results based on image analysis sound really interesting, however, the authors should improve the paper mainly about the description of computed tomography methodology. The description of procedures, that combines CT and enzymatic analyses, also needs special attention. One of my main concerns is how the authors converted the 3D volume of pores information, obtained via CT, for the equivalent cylindrical diameter or pore radius studied? Was this procedure based on the capillary equation? How could it be done? Were the 3D pores considered as equant shaped? I made several comments to the authors, which can be viewed in the pdf file attached.

Reviewer: How the authors converted the 3D volume of pores information, obtained via CT, for the equivalent cylindrical diameter or pore radius studied?

Reviewer I. 72 Equivalent cylindrical diameter?

Reviewer, line 85: As microtomography and image analysis are important for your work, a brief paragraph about the state of art of these techniques could improve the introduction.

Reviewer I 89 – Diameter or pore radius?

Reviewer I. 96: Some presentation in the introduction about the function of these pores could improve a lot the paper!

Reviewer I. 108 – What is the difference between bulk and intact soil samples?

Reviewer I. 147 Measured by using each method? Soil moisture content at saturation? The relation between the soil bulk and particle densities? CT? Please, give details in the methodology! Any information about micro, meso and macropores? The results of these pore types can improve the discussion based on pore functions!

Reviewer I.149 Crop rotation? Agricultural traffic in the area?

Reviewer I.153 Why this value was selected as threshold?

Reviewer Fig. 2a Threshold procedure? Perhaps, you can also include in this graph the porosity measured by CT! Which procedure was followed for separating the roots of the remaining soil matrix and pores in the segmentation procedure? Were the roots considered as pores?

Reviewer Fig. 2b What these numbers mean? How they were obtained?

Reviewer Fig. 4 Was this pore size accessed by CT? This is not clear to me? 2D or 3D analysis? – about pore sizes.

Reviewer I. 247 Which CT parameters? Circularity? Perimeter? Surface area? – about pore architecture, need to define what it is

Reviewer I. 247-248 Was the segmentation procedure the same for all the treatments? Any importance of differences in the root systems?

Reviewer I. 340 Mainly when we have 2:1 clays!

Reviewer I.394 Size of the cores? Details about the sampling procedure? Height, diameter, etc.! Sampling procedure can affect the quality of samples!

Reviewer I.394 Why 5-10 cm? Normally, it is hard to collect sample exactly in this specific layer!

Reviewer I. 398: Which procedure was followed to maintain the structure of the samples prior shipping? Any image analysis of the one of the samples before and after shipping?

Reviewer I. 398: Do you have information about the soil bulk density? This type of information is important for the image segmentation process!

Reviewer I. 401. Were the samples dried before CT analysis? How the soil water content inside the samples was managed before segmentation? This could be a problem mainly with you have soil bulk density values close to the water density value!

Reviewer I. 401: v-tomex- s, m or l?

Reviewer I. 406. Current. The electrical flux is related to the number electric line forces passing through a surface.

Reviewer I. 408. Any beam hardening problem? Were the sample scanned inside any container?

Reviewer I. 409. Which method? FB?

Reviewer I. 410. Please, give more details about the ROI selected for these analyses as well as the voxel size obtained!

Reviewer I.412 Please, give details about the parameter values utilized in this procedure!

Reviewer I. 414 Because of this, the information about the sample size is crucial! This is also important to analyze questions related to REV!

Reviewer I. 414: This information is very important to guarantee the quality of the analysis based on microtomography. Please, give give more details about this procedure. This is also important for any one that would like to reproduce your study or procedures taken!

Reviewer I. 415: 8, 16, 32 bits? Please, give details!

Reviewer I. 416 I do not know this plugin, but the psd obtained is based on ecd or volume?

Reviewer I. 417: Based on each pore shape? Sphere, cylinder, cube? Each voxel size?

Reviewer I. 421: It is hard to observe it in the figure 3c! Please, give more details about the procedures to obtain the pore sizes. Capillary equation? If so, how the authors managed the situations in which the pores have prolate, oblate, etc. shapes? Normally, the soil pores are very complex and connected, they are composed of several voxels “aligned” in specific directions. To consider these voxels as equant shaped ones, most of time, can lead us to wrong conclusions about pore morphology.

Reviewer I. 421 What for? Please, give details!

Reviewer I. 441 Was this procedure followed to calculate the gravimetric water content? Normally samples are dried in an oven at 105 C!

Reviewer I. 457 ? Sorry for ignorance!

Reviewer I. 461. Any supplementary illustration for this experimental procedure?

Reviewer I. 472. How many soil slices were studied?

Reviewer I. 475 Are the samples dried for this procedure?

Reviewer I. 480. To be honest, this figure 3b is not necessary in my opinion. Perhaps, an schematic drawing presenting the procedures for this analysis could be more interesting!

Reviewer I.486 early...

Reviewer I. 493. Please, give details? Any possibility of alterations in the soil structure or pores be filled by disperse clay, for example? Or these are not important for these analyses?

Reviewer I. 497. Any image to highlight this procedure? Perhaps, as a supplementary figure!

Reviewer I.500 Were the same samples analyzed? Were these procedures (zymograms and CT analyses) carried out at the same lab?

Reviewer I.505. Was the same resolution obtained in each technique?

Reviewer I.511. Any details about the soil composition (texture, mineralogy, etc.) among cropping systems, which can affect the results of this procedure?

Reviewer I. 514: EN or EPC number? Which procedure? Plugins, methods, etc... I did not find any discussion or results presentation about the connectivity. Certainly, the authors would obtain benefits from this parameter due to differences in the root system of the cultures analyzed!

Reviewers' comments:

Reviewer #4 (Remarks to the Author):

I read the paper carefully and made several comments on it. I believe that this research work is important and present interesting results about the soil porous system and soil carbon stabilization. The results based on image analysis sound really interesting, however, the authors should improve the paper mainly about the description of computed tomography methodology. The description of procedures, that combines CT and enzymatic analyses, also needs special attention. One of my main concerns is how the authors converted the 3D volume of pores information, obtained via CT, for the equivalent cylindrical diameter or pore radius studied? Was this procedure based on the capillary equation? How could it be done? Were the 3D pores considered as equant shaped? I made several comments to the authors, which can be viewed in the pdf file attached.

Response: We have responded to all Reviewer's comments and provided clarifications on all requested accounts. Please see below the point-by-point list of all reviewer's comments made in the manuscript's pdf file with the manuscript's line numbers provided and the line numbers for the relevant modifications/edits made in the revised manuscript. We are grateful for the Reviewer's efforts, which we believe, significantly improved clarity and quality of our manuscript.

Please note that we had to be mindful of Nature Comm. format, thus could only sparingly increase the size of the Introduction and Results sections. However, we provided all requested details in the Materials and Methods section.

Reviewer: How the authors converted the 3D volume of pores information, obtained via CT, for the equivalent cylindrical diameter or pore radius studied?

Response: The pore size distribution was obtained using Xlib plugin for ImageJ as described in details in Münch and Holzer (2008). We added the following description to the Methods: "*Pore size distributions (PSD) were obtained using the Xlib plugin for ImageJ⁸⁴. We used the continuous 3D PSD option of the software, which provides radii of the largest spheres, that fit into the 3D pore volume, as described in detail in ref. ⁸⁴. Therefore, the pore size at a specific location was defined as the radius of the maximally inscribable sphere at this location. The ratio of the number of voxels occupied by pores of each size class and the total number of the voxels within the soil core was used to report the pore-size distribution results (Fig. 2b).*" (l. 450-455 in the revised manuscript).

Reviewer I. 72 Equivalent cylindrical diameter?

Response: The pore size at a specific location was defined as a radius of the maximally inscribable sphere (see previous response).

Reviewer, line 85: As microtomography and image analysis are important for your work, a brief paragraph about the state of art of these techniques could improve the introduction.

Response: We have added the following sentence and two relevant references highlighting state-of-the-art in μ CT use: *“To test this hypothesis we explored the soils’ pore characteristics via X-ray computed micro-tomography (μ CT). Non-invasive μ CT scanning supplies detailed 3D information on the material density distribution within a sample. Hence it is very well suited to outline the locations and geometry of pores. This in turn improves our understanding of chemical and biological processes taking place within the soil pore space^{46, 47}.“* (l. 82-86 in the revised manuscript).

Reviewer I 89 – Diameter or pore radius?

Response: Here and everywhere else in the manuscript we refer to pore sizes in terms of their radii. We have clarified that on l. 73, l. 93 and in relevant figure captions in the revised manuscript.

Reviewer I. 96: Some presentation in the introduction about the function of these pores could improve a lot the paper!

Response: We have added the following to the introduction on the functioning of these pores: *“Pores in the 30-150 μ m radius size range may be especially important³⁷ because they function as likely routes of O₂ influx^{38, 39} and localities of new C inputs from fine plant roots⁴⁰. Their special role is evidenced by greater microbial activity⁴¹⁻⁴⁴, higher abundance of diverse taxa⁴⁵, and presence of dissolved organic matter enriched in lipids and depleted in lignin^{46, 47}.“* (l. 72-76 in the revised manuscript).

Reviewer I. 108 – What is the difference between bulk and intact soil samples?

Response: We have clarified this as follows: *“We collected disturbed soil samples and undisturbed samples, referred to as intact cores, at a replicated field experiment in southwest Michigan, USA.”* (l. 112-114 in the revised manuscript).

Reviewer: Increase font size on all graphs

Response: We have increased the font size as requested.

Reviewer I. 147 Measured by using each method? Soil moisture content at saturation? The relation between the soil bulk and particle densities? CT? Please, give details in the methodology! Any information about micro, meso and macropores? The results of these pore types can improve the discussion based on pore functions!

Response: We have provided the following information on total porosity determination: “*The total porosity of each intact soil sample was calculated from its soil volume determined from the μ CT image, gravimetric soil moisture content measured at the end of the study, and soil weight. A particle density of 2.6 g cm^{-3} was assumed for the total porosity calculations.*” (l. 476-479 in the revised manuscript).

Please note that we did not classify the pores into micro-, meso-, and macro- categories. Instead, we just reported the data on specific pore size groups.

Reviewer I.149 Crop rotation? Agricultural traffic in the area?

Response: We have provided the following description of the studied cropping systems in the Methods section: “*The five studied systems were: continuous corn and continuous corn with winter cover crop of cereal rye, a monoculture switchgrass, a hybrid poplar (*Populus nigra* \times *P. maximowiczii* ‘NM6’) with herbaceous understory¹², and an early successional community abandoned from agriculture in 2008. The experimental site was plowed prior to establishment of the cropping systems after which no further plowing took place in either of the systems. The two continuous corn systems were managed as no-till. All systems were managed using local best agronomic practices⁴⁸.*” (l. 402-408 in the revised manuscript).

Please note that the soil sampling sites were located in the central portions of the experimental plots and experienced agricultural traffic typical to each system.

Reviewer I.153 Why this value was selected as threshold?

Response: In order to not excessively increase the size of the Results section we have provided the justification in the Methods: “*As a measure of the soil matrix fraction that was potentially affected by C processing taking place within the imageable pores ($\geq 30 \mu\text{m}$), we used the soil matrix volume located within $180 \mu\text{m}$ distance from these pores (Fig. 3c). Detailed descriptions of the volume determination and experimental justification for its selection is provided in ref⁸¹. Briefly, the volume was determined via a series of 3D dilations using ImageJ until $\sim 180 \mu\text{m}$ distance from the pore surface was reached. The value of $180 \mu\text{m}$ was selected as consistent with distances for spatial correlations in microbial colony distributions⁵³ and spatial patterns of soil C distributions⁵⁴ in previous micro-scale studies of soil matrix.*” (l. 467-474 in the revised manuscript).

Reviewer Fig. 2a Threshold procedure? Perhaps, you can also include in this graph the porosity measured by CT! Which procedure was followed for separating the roots of the remaining soil matrix and pores in the segmentation procedure? Were the roots considered as pores?

Response: We have provided the following description in the Methods: “*Particulate organic matter (POM), including plant roots, on the images was identified as described in ref⁷⁹. Briefly, we, first, visually determined minimum and maximum gray scale values for initial POM/root thresholding. Then, the initial thresholding was performed and a set of 3D erosion/dilation steps was applied to eliminate partial volume artefacts. A 3D Gaussian filter was applied in order to*

restore the size of the POM/root fragments after the removal of the partial volume voxels. Then, the image was binarized once more and the Particle Analyser plugin was used to select only POM fragments with volume exceeding 0.016 mm^3 . The analyses were performed using BoneJ plug-in of Image J. In further analyses POM/roots were considered separately from pore and solid fractions." (1.459-466 in the revised manuscript).

Reviewer Fig. 2b What these numbers mean? How they were obtained?

Response: We have added the following explanation to the Methods section, describing the thresholding and pore-size distribution determination method: "*Then, pore size distributions (PSD) were obtained using the Xlib plugin for ImageJ⁸⁴. We used the continuous 3D PSD option of the software, which provides radii of the largest spheres, that fit into the 3D pore volume, as described in detail in ref. ⁸⁴. Therefore, the pore size at a specific location was defined as the radius of the maximally inscribable sphere at this location. The ratio of the number of voxels occupied by pores of each size class and the total number of the voxels within the soil core was used to report the pore-size distribution results (Fig. 2b).*" (1.450-455 in the revised manuscript).

We have also added the following to the caption of Fig. 2b to clarify: "*The μCT -derived pore size fractions are shown in volume percents of the total soil sample volumes.*" (1. 176-178 in the revised manuscript).

We have now also added the units to Fig. 2b, as requested.

Reviewer Fig. 4 Was this pore size accessed by CT? This is not clear to me? 2D or 3D analysis? – about pore sizes.

Response: We provided the following description in the Methods: "*Porosity $< 30 \mu\text{m}$ was determined as the difference between the total porosity and the image-based porosity, that is, the volume of pores with radii $> 30 \mu\text{m}$, obtained from the images.*" (1. 479-481 in the revised manuscript).

Reviewer I. 247 Which CT parameters? Circularity? Perimeter? Surface area? – about pore architecture, need to define what it is

Response: We have rephrased as following: "*Nine years of implementing cropping systems with different plant species diversities led to the formation of contrasting soil pore size distributions*" (1. 236-237 in the revised manuscript).

Reviewer I. 247-248 Was the segmentation procedure the same for all the treatments? Any importance of differences in the root systems?

Response: We have added the following to the Methods: "*The same image processing procedures were applied to soil samples from all studied systems.*" (l. 436-437 in the revised manuscript).

Reviewer I. 340 Mainly when we have 2:1 clays!

Response: We have added the following to the Discussion: "*Systems where soils experience greater drying in summer would be expected to have a greater presence of large pores (cracks) due to soil shrinkage. This phenomenon is commonly observed in soils with high clay contents (2:1), but still, though to a lesser extent, is present in sandy loam soils*". (l. 353-356 in the revised manuscript).

Reviewer I.394 Size of the cores? Details about the sampling procedure? Height, diameter, etc.! Sampling procedure can affect the quality of samples!

Response: We have provided the following details on the soil sampling in the Methods: "*Two intact soil cores (5 cm Ø x 5 cm height) were collected from 4 replicated plots of each bioenergy system for a total of 40 cores in early spring 2017. The soil cores were taken using a soil core sampler (Soil Moisture Equipment Corp.) into an acrylic cylinder located within the sampler. The sampler had two rings, 5-cm height each, and soil from the first ring was discarded. Thus, the cores were collected precisely from 5-10 cm depth within the soil profile. A disturbed soil sample and an additional soil core for bulk density measurement were also collected from each sampling location.*" (l. 409-415 in the revised manuscript).

Reviewer I.394 Why 5-10 cm? Normally, it is hard to collect sample exactly in this specific layer!

Response: This is the layer were we expected the differences between the studied systems to be most pronounced. Moreover, all studied systems were no-till, thus collecting soil samples from 5-10 cm depth was straightforward.

Reviewer I. 398: Which procedure was followed to maintain the structure of the samples prior shipping? Any image analysis of the one of the samples before and after shipping?

Response: We have provided the following details on the soil preparation to shipping in the Methods: "*To preserve the cores during shipping, each core was tightly closed on both ends with ridged foil caps and wrapped in several layers of plastic using duct tape.*" (l. 419-421 in the revised manuscript).

Unfortunately, we did not have the scanning capabilities to scan the cores prior to shipping them. This is why they had to be shipped to the lab that had the scanner. However,

based on the visual assessment of the cores before and after shipment their structure was not affected.

Reviewer I. 398: Do you have information about the soil bulk density? This type of information is important for the image segmentation process!

Response: We have provided the following details in the Methods: “*A disturbed soil sample and an additional soil core for bulk density measurement were also collected from each sampling location. Bulk density was measured for each sampling location using the core method^{s0} and is reported in ref^{s1}.*” (l. 413-416 in the revised manuscript)

Reviewer I. 401. Were the samples dried before CT analysis? How the soil water content inside the samples was managed before segmentation? This could be a problem mainly with you have soil bulk density values close to the water density value!

Response: Soil cores were wrapped with the foil and kept in the cold room prior X-ray CT scanning. The soil bulk density ranged from 1.51 to 1.77 g/cm³ in the scanned cores, which by far exceeded density of water. Therefore, presence of water did not create a problem for image segmentation.

Reviewer I. 401: v-tomex- s, m or l?

Response: We have provided the following information in the Methods: “*Intact soil cores were subjected to X-ray scanning using a GE Phoenix v|tome|x m scanner at the Institute of Soil and Environment at the Swedish University of Agricultural Sciences in Uppsala. The X-ray scanner was equipped with a 240 kV tube, a tungsten target and a 16'' flat panel detector with 2014 × 2014 detector crystals (GE DRX250RT).*” (lines 423-427 in the revised manuscript).

Reviewer I. 406. Current. The electrical flux is related to the number electric line forces passing through a surface.

Response: We have replaced “flux” with “current” as suggested.

Reviewer I. 408. Any beam hardening problem? Were the sample scanned inside any container?

Response: We have provided the following details: “*Soil cores were scanned inside their acrylic rings. Beam hardening artefacts were not observed.*” (l. 431-432 in the revised manuscript).

Reviewer I. 409. Which method? FB?

Response: We have now modified the respective sentence as follows: “3D μ CT X-ray images were reconstructed using the filtered back-projection approach implemented in GE software *datos|x*” (l. 432-433 in the revised manuscript).

Reviewer I. 410. Please, give more details about the ROI selected for these analyses as well as the voxel size obtained!

Response: We have provided the following details: “*The entire soil core, including the acrylic cylinder, was scanned. Each image had a voxel size of 29 μ m in x, y and z direction.*” (l. 433-434 in the revised manuscript).

Reviewer I.412 Please, give details about the parameter values utilized in this procedure!

Response: We have provided the following details in the Methods: “*Preprocessing consisted of a 3D median filter with a radius of two voxels in all directions using Median 3D filter tool from ImageJ.*” (l. 437-439 in the revised manuscript).

Reviewer I. 414 Because of this, the information about the sample size is crucial! This is also important to analyze questions related to REV!

Response: We have provided all requested sample size information as described earlier.

Reviewer I. 414: This information is very important to guarantee the quality of the analysis based on microtomography. Please, give give more details about this procedure. This is also important for any one that would like to reproduce your study or procedures taken!

Response: We have provided the following details on the thresholding procedure in the Methods: “*For segmenting the images into pores and non-pores we used the minimum error approach⁸² on the respective image histograms (8-bit. The grayscale histograms (8-bit) of all soil cores exhibited a two-distribution pattern, with one distribution corresponding to air+liquid and the other to solid portions of the images. Thus, following ref.⁸³, the two-Gaussian fits were applied to the histograms. The threshold was computed as a greyscale value that minimized the difference between the overlapping areas of the two distributions. This approach conserves the voxel balance between the two segmented phases (solid and air+liquid). The necessary computations were performed in the Regression Wizard tool of the SigmaPlot software (Systat Software, Inc). The segmentations were conducted separately for each soil core.*” (l. 441-449 in the revised manuscript).

Reviewer I. 415: 8, 16, 32 bits? Please, give details!

Response: 8-bit images were used for these analyses. The information has been added to the revised manuscript l. 442, as detailed in the previous response.

Reviewer I. 416 I do not know this plugin, but the psd obtained is based on ecd or volume?

Response: The plug-in uses the maximum inscribed sphere approach and is volume based. We have added that information to the revised manuscript (l. 450-457 in the revised manuscript).

Reviewer I. 417: Based on each pore shape? Sphere, cylinder, cube? Each voxel size?

Response: Sphere-based. We have added that information to the revised manuscript (l. 450-457 in the revised manuscript).

Reviewer I. 421: It is hard to observe it in the figure 3c! Please, give more details about the procedures to obtain the pore sizes. Capillary equation? If so, how the authors managed the situations in which the pores have prolate, oblate, etc. shapes? Normally, the soil pores are very complex and connected, they are composed of several voxels “aligned” in specific directions. To consider these voxels as equant shaped ones, most of time, can lead us to wrong conclusions about pore morphology.

Response: We did not use the capillary equation approach, the detailed description of the procedures is now added to the revised manuscript (l. 450-457). We do agree that definition of the pore size affects pore morphology characteristics. Since most pore water exists in soil in the form of menisci, we assume that continuous 3D PSD calculations based on inscribed spheres provide the most accurate estimates of pore necks and water distribution in soil

Reviewer I. 421 What for? Please, give details!

Response: We conducted the dilations to quantify the soil volume in vicinity of $>30 \mu\text{m}$ pores. As mentioned earlier, more details are now provided in the revised manuscript.

Reviewer I. 441 Was this procedure followed to calculate the gravimetric water content? Normally samples are dried in an oven at 105 C!

Response: After gravimetric water content analysis the soil samples were used for total C and C/N analyses. Therefore, lower drying temperature but longer drying period were used (standardized protocols of Long Term Ecological Research network <https://lter.kbs.msu.edu/protocols/24>)

Reviewer I. 457 ? Sorry for ignorance!

Response: *aka* is an abbreviation for “also known as”. To avoid confusion we replaced it with “referred to as”.

Reviewer I. 461. Any supplementary illustration for this experimental procedure?

Response: We have added a figure illustrating the set up to the supplementary materials (Supplementary Fig. 6a) and a schematic outline of the sample processing (Supplementary Fig. 6b).

Reviewer I. 472. How many soil slices were studied?

Response: We have clarified as following in the Methods: "*A total of 13 soil cores (2-3 cores per cropping system) were subjected to zymography analysis, with 10-16 zymograms obtained from each core for a total of 150 enzyme maps, one map per each soil slice.*" (1.525-527 in the revised manuscript)

Reviewer I. 475 Are the samples dried for this procedure?

Response: We have added the following explanation to the Methods: "*The soil moisture levels were not manipulated prior to the measurements.*" (l. 512-513 in the revised manuscript).

Reviewer I. 480. To be honest, this figure 3b is not necessary in my opinion. Perhaps, an schematic drawing presenting the procedures for this analysis could be more interesting!

Response: Since in Nature Comm.'s format a reader will only get to the Methods description after the Results section, we believe it is helpful to provide a general idea of how the data were obtained. We believe Fig. 3b serves that purpose, thus we would prefer to keep it. However, we have also added a schematic drawing of the procedure, as requested by the reviewer (Supplement Fig. 6b in the revised manuscript).

Reviewer I.486 early...

Response: Corrected here and in the rest of similar instances through the manuscript.

Reviewer I. 493. Please, give details? Any possibility of alterations in the soil structure or pores be filled by disperse clay, for example? Or these are not important for these analyses?

Response: The weight of the sandbag was only ~20 g, it could not possibly have any effect on soil structure or pores. We have now specified the weight of the bag in the revised manuscript.

Reviewer I. 497. Any image to highlight this procedure? Perhaps, as a supplementary figure!

Response: We have added supplementary figure describing the procedure, as requested (Supplementary Fig. 6a and b in the revised manuscript).

Reviewer I.500 Were the same samples analyzed? Were these procedures (zymograms and CT analyses) carried out at the same lab?

Response: That is correct, the same samples were used in both zymography and CT analyses. First, each sample was subjected to X-ray CT scanning, then it was subjected to zymography. Hopefully, the added schematic diagram clarifies this now.

Reviewer I.505. Was the same resolution obtained in each technique?

Response: The voxel edge length for X-ray CT images was equal to 29 μm (l. 434 in the revised manuscript). The resolution of the zymography maps was estimated to be $\sim 100 \mu\text{m}$, but cannot be determined exactly. In order to compare the X-ray CT and zymography data, we aggregated the image data to voxel and pixel sizes of 1 mm^3 and 1 mm^2 , respectively (l. 560-563 in the revised manuscript).

Reviewer I.511. Any details about the soil composition (texture, mineralogy, etc.) among cropping systems, which can affect the results of this procedure?

Response: We have added the information on sand, silt, clay percentages of the studied soil (l. 400 in the revised manuscript). The cropping systems were randomly assigned to the experimental field plots within the same experimental site, thus no significant differences among the systems in terms of texture and mineralogy were expected. And no significant differences were observed when the texture data of the studied systems were compared.

Reviewer I. 514: EN or EPC number? Which procedure? Plugins, methods, etc... I did not find any discussion or results presentation about the connectivity. Certainly, the authors would obtain benefits from this parameter due to differences in the root system of the cultures analyzed!

Response: We apologize for the confusion. During previous revision the connected pore results were excluded from this manuscript. Thus, the mention of connected pores here is in error and was deleted from the revised manuscript.

REVIEWERS' COMMENTS:

Reviewer #4 (Remarks to the Author):

I believe that after all the changes made by the authors, in my opinion, the paper can be finally accepted for publication. I have no further comments regarding the content of the manuscript.